# Precipitation response to aerosol-radiation and aerosol-cloud interactions in regional climate simulations over Europe

José María López-Romero[1], Juan Pedro Montávez[1], Sonia Jerez[1], Raquel Lorente-Plazas[1,2], Laura Palacios-Peña[1], and Pedro Jiménez-Guerrero[1,3]

[1]Physics of the Earth, Regional Campus of International Excellence (CEIR) "Campus Mare Nostrum", University of Murcia, 30100 Murcia, Spain
[2]Department of Meteorology, Meteored, 30893, Murcia, Spain
[3]Biomedical Research Institute of Murcia (IMIB-Arrixaca), 30120 Murcia, Spain

**Correspondence:** Juan Pedro Montávez (montavez@um.es)

**Abstract.**

The effect of aerosols on regional climate simulations presents large uncertainties due to their complex and non-linear interactions with a wide variety of factors, including aerosol-radiation (ARI) and aerosol-cloud (ACI) interactions. These interactions are strongly conditioned by the meteorological situation and type of aerosol but, despite their increase, only a limited number of studies have covered this topic from a regional and climatic perspective.

This contribution thus aims to quantify the impacts on precipitation of the inclusion of ARI and ACI processes in regional climate simulations driven by ERA20C reanalysis. A series of regional climatic simulations (for the period 1991-2010) for the Euro-CORDEX domain were conducted including ARI and ARI+ACI (ARCI), establishing as a reference a simulation where aerosols were not included interactively (BASE).

The results show that the effects of ARI and ACI on time-mean spatially averaged precipitation over the whole domain are limited. However, a spatial redistribution of precipitation occurs when the ARI and ACI processes are introduced in the model, as well as changes in the precipitation intensity regimes. The main differences with respect to the base-case simulations occur in central Europe, where a decrease in precipitation is associated with a depletion in the number of rainy days and clouds at low level (CLL) . This reduction in precipitation presents a strong correlation with the ratio PM2.5/PM10, since the decrease is especially intense during those events with high values of that ratio (pointing to high levels of anthropogenic aerosols) over central Europe. The precipitation decrease occurs for all ranges of precipitation rates. On the other hand, the model produces an increase in precipitation over the eastern Mediterranean basin associated with an increase of clouds and rainy days when ACI are implemented. Here, the change is caused by the high presence of PM10 (low PM2.5/PM10 ratios, pointing to natural aerosols). In this case, the higher amount of precipitation affects only days with low rates of precipitation. Finally, there are some disperse areas where the inclusion of aerosols leads to an increase in precipitation, especially for moderate and high precipitation rates.

# 1 Introduction

The importance of atmospheric aerosols has multiple aspects, all of great scientific and socioeconomic relevance. First, the World Health Organization (WHO, 2013) has recognized that the degradation of air quality by atmospheric aerosols is a threat to human health. Second, the Fifth Assessment Report (AR5) of the Intergovernmental Panel on Climate Change (IPCC) points to atmospheric aerosols as one of the main sources of uncertainty in current climate simulations (Boucher et al., 2013). Myhre et al. (2013) indicate that the uncertainty in the radiative forcing produced by aerosols greatly exceeds that of all other forcing mechanisms combined.

Despite the increasing number of articles published on the interactions between aerosols and climate over the last 20 years (Fuzzi et al., 2015), the uncertainty associated with the estimated radiative forcing attributed to the interactions between aerosols and clouds has not diminished during the last four cycles of the IPCC (Seinfeld et al., 2016). One of the main tools for estimating the impact of atmospheric aerosols on climate is the use of global and regional climate models (Boucher et al., 2013). However, many of the simulations attempting to reproduce both the present climate and future climatic scenarios, or the extreme events that occur in situations of present or future climates, do not take into account the role of aerosol-radiation and aerosol-cloud interactions (ARI and ACI, respectively, in the terminology of AR5).

In addition to their radiative effect, aerosols act as condensation nuclei for cloud formation and can therefore affect precipitation in several ways (Andreae and Rosenfeld, 2008; Rosenfeld et al., 2008). Rosenfeld et al. (2008) studied the role of aerosols in polluted and pristine atmospheres for tropical areas. In polluted atmospheres, as there is a larger amount of condensation nuclei for the same humidity, the cloud drops are smaller and therefore aerosols hamper precipitation. The slower cloud-droplet-to-rain conversion allows the droplets to be transported above freezing level and thus the latent heat released in freezing intensifies the convection. However, this has no general validity, since this behavior could change locally, depending on the area. Indeed, understanding and characterizing the role that aerosols play in the development of convective clouds is today a cutting-edge scientific challenge (Archer-Nicholls et al., 2016). Authors such as Seifert et al. (2012); Fan et al. (2013) find a very weak effect on precipitation by introducing aerosol-cloud interactions. Da Silva et al. (2018) analyze the effects on microphysics for the year 2013 for the Euro-Mediterranean region and conclude that precipitation decreases when there is a higher amount of aerosols.

Better understanding of the ARI and ACI interactions is therefore essential for identifying climate change and its manifestation through changes in the frequency and severity of precipitation events (Huang et al., 2007; Khain et al., 2008; Stevens and Feingold, 2009; Fuzzi et al., 2015). Along the same lines, works such as Shrivastava et al. (2013); Forkel et al. (2015); Turnock et al. (2015); Yahya et al. (2016); Palacios-Peña et al. (2018, 2019); Pavlidis et al. (2020) highlight that it is necessary to use regional climate/chemical coupled models to investigate ACI interactions in more detail. These studies cover mainly the continental US, Asia and Europe and investigate chemical and meteorological variables such as precipitation, temperature and radiation. As indicated by Seinfeld et al. (2016), improving the estimation of the aerosol impact on clouds and reducing associated uncertainty are critical challenges for climate modeling studies. Despite the errors and uncertainties related to the role of aerosols in the climate system (Jiménez-Guerrero et al., 2013), only a small number of scientific papers have considered

the analysis of climatic events using simulations that include ARI and ACI interactions, which may strongly condition the representation and definition of events associated with precipitation and cloudiness (Prein et al., 2015; Baró et al., 2018).

In regional climate models, representation of the radiative effect of aerosols (ARI) is traditionally established by a constant aerosol optical thickness (AOD) value and a predetermined and abundant number of cloud condensation nuclei (CCN) (Forkel et al., 2015). Although the lack of CCN is hardly ever a limiting factor for cloud formation (this could perhaps happen in remote marine locations in very specific conditions), a low CCN value may result in clouds that precipitate more readily, which can reduce the cloud lifetime and therefore the average cloud fraction (Stevens and Feingold, 2009). To obtain a more realistic model, ARI and ACI interactions, which require models in which meteorology-climatology, radiation, clouds and aerosol atmospheric chemistry are coupled in a fully interactive way, must be included in the simulation (Grell and Baklanov, 2011; Baklanov et al., 2014). Fully coupled climate-chemistry models (*on-line*) make it possible to explain the feedback mechanisms between simulated aerosol concentrations and meteorological variables.

In simulations including ARI, the number of CCN remains unchanged, but the concentration of aerosols and their impact on the radiative balance is dynamically modeled (Houghton et al., 2001; Andreae et al., 2005). A region with a high emission of black carbon will absorb more radiation and increase the temperature of that layer of the atmosphere, favoring the destruction of clouds. However, an area with emissions of clear natural aerosols (e.g. sea salt) will favor radiative cooling due to the scattering of radiation (Yu et al., 2006).

Also, a further refinement in the configuration of the model adds the aerosol-cloud interactions. In this case, on-line estimation of aerosol concentrations is conducted in each timestep of the model (as in the previous case), but this dynamical estimation is used to both calculate the radiative budget (as in ARI) and to estimate CCN for cloud formation. This will affect the number of drops within the cloud as well as their size, modifying the cloud's optical properties and thus its radiative balance (Twomey, 1977), and whether they reach the critical size to precipitate or not (Rosenfeld et al., 2008).

Introducing ACI interactions adds a level of complexity that brings the model configuration closer to real processes. However, it has a great computational cost and can increase calculation times from six- to ten-fold (López-Romero et al., 2016; Palacios-Peña et al., 2020). It is therefore reasonable that most of the studies carried out so far with regional models taking into account these interactions have been for episodical case studies (Yang et al., 2012; Brunner et al., 2015; Palacios-Peña et al., 2019) and only a very limited number of contributions cover climatic periods with a general analysis (e.g. Witha et al. (2019); Pavlidis et al. (2020)).

Hence, in this work, the role of ARI and ACI on precipitation and cloudiness over Europe has been exhaustively explored. For this purpose, regional climate simulations (1991-2010) for the Euro-CORDEX (Jacob et al., 2014) domain were carried out with WRF-Chem in order to account for the influence of atmospheric aerosols on the aforementioned variables.

## 2 Data and Methods

### 2.1 Experimental setup

Regional climate simulations were carried out using the WRF-Chem model (v.3.6.1), both uncoupled from chemistry (WRF stand-alone configuration, Skamarock et al. (2008)) and including a full on-line coupling with atmospheric chemistry and pollutant transport (for including ARI and ACI processes) (Grell et al., 2005).

Three different experiments were performed in this contribution. The first, BASE, consists in prescribing AOD and CCN, excluding ARI and ACI interactions. The second, ARI, includes only Aerosol Radiation Interactions (direct and semidirect effects). The third, ARCI, includes both aerosol-radiation and aerosol-cloud interactions (direct, semidirect and indirect effects). In ARI and ARCI, aerosols are calculated online. These experiments allow the effects of the aerosols on clouds and precipitation from a climatic perspective to be disentangled.

In the BASE experiment, aerosols are not treated interactively, but by using the default WRF configuration, which considers 250 CCN per $cm^3$ and setting AOD to 0. In the ARI experiment, aerosols are treated online and ARI processes are activated in the model (Fast et al., 2006), but CCN remain as in the stand-alone version. The ARCI experiment includes the aforementioned ARI and, in addition, permits aerosols to interact with the microphysics processes. The description of ARCI as implemented in the simulations can be found in Palacios-Peña et al. (2020), as can validation of the AOD fields. To summarize, ARCI in WRF-Chem were implemented by linking the simulated cloud droplet number with the Lin microphysics scheme (Lin et al., 1983) , turning it into a two-moment scheme. Therefore, the droplet number affects both the calculated droplet mean radius and the cloud optical depth (Chapman et al., 2009).

The spatial configuration consists of two unidirectionally-nested domains (one-way nesting).The domains used are shown in Figure 1). The inner domain is compliant with Euro-Cordex recommendations (Jacob et al., 2014). It covers Europe with a spatial resolution of 0.44º in latitude and longitude ($\sim$ 50km). The outer domain has a spatial resolution of about 150km and extends southward to a latitude of approximately 20°N. The design of this domain aims to cover the most important dust emission areas of the Saharan desert (Goudie and Middleton, 2001; Middleton and Goudie, 2001; Rodrıguez et al., 2001; Goudie and Middleton, 2006) that are introduced to the inner domain through boundary conditions (Palacios-Peña et al., 2019). Nudging was used for the outer domain so that the atmospheric dynamics did not significantly vary (Liu et al., 2012). In the vertical, 29 non-uniform sigma levels were used, with higher density levels near the surface. The upper limit was set at the 50 hPa level.

The design of the physical configuration of the model was based on the compatibility with the chemical module and previous works (Baró et al., 2015; Palacios-Peña et al., 2016; Baró et al., 2017; Palacios-Peña et al., 2017, 2019). In addition to microphysics (Lin scheme), another important parameterization concerns radiation. The interactions of aerosol and clouds with incoming solar radiation were implemented by linking the simulated cloud droplet number with the RRTMG scheme and Lin microphysics (further details in Palacios-Peña et al. (2020)). Therefore, the droplet number will affect both the calculated droplet mean radius and cloud optical depth. This should allow the dynamical treatment of aerosols and greenhouse gases in order to estimate the radiative budget. The RRTMG radiative scheme was used for both long and short wave (Iacono et al.,

2008), while the Grell 3D scheme was used for the cumulus parameterization, (Grell, 1993; Grell and Devenyi, 2002). The boundary layer was modelled with the Yonsei University scheme (Hong et al., 2006). The surface layer was parameterized using the Jiménez et al. (2012) scheme. Finally, the NOAH model (Tewari et al., 2004) was the land-soil model chosen to simulate the land-atmosphere interactions.

As mentioned above, aerosols are treated on-line, i.e. the model uses changing aerosols originating from anthropogenic emissions and generating natural aerosols throughout the interaction between atmospheric conditions and surface properties. Regarding the configuration and treatment of aerosols and gases, the gas-phase chemical mechanism RACM-KPP was used (Stockwell et al., 2001; Geiger et al., 2003) coupled to the GOCART aerosol scheme (Ginoux et al., 2001a; Chin et al., 2002). The photolysis module Fast-J (Wild et al., 2000) was used to feed photochemical reactions. Biogenic emissions were calculated
online using the Model of Emissions of Gases and Aerosols from Nature model (MEGAN) (Guenther et al., 2006). Dust and marine spray were simulated with GOCART (Ginoux et al., 2001b; Chin et al., 2002). Simulated aerosols included five species: sulfate, mineral dust, sea salt, organic matter and black carbon. Anthropogenic emissions were taken from the Intercomparison Project of Atmospheric and Climate Chemistry Models (Lamarque et al., 2013) and remained unchanged during the simulation period (monthly values for 2010). The ability of this configuration to represent the Aerosol Optical Depth has been already
extensively evaluated in Palacios-Peña et al. (2020). More details about the treatment of aerosols and their interactions can be found in Jerez et al. (2020b). The mean fields of these aerosols as well as the AOD are presented as supplementary material (Figures S1-S5).

The simulated historical period for the three simulations covers the two decades from 1991 to 2010. The boundary and initial conditions were extracted from the ECMWF reanalysis ERA20C (ECMWF, 2014; Hersbach et al., 2015), which has a
140 horizontal resolution of approximately 125 km (T159). The simulations were run splitting the full period into sub-periods of 5 years with a spin-up period of 4 months, then beginning with the direct interpolation of the soil data of the reanalysis. After removing the spin-up period, which was chosen in accordance with the results of Jerez et al. (2020a), the model outputs were merged. This methodology was tested in Jerez et al. (2020a). The boundary conditions for the outer domain were updated every 6 hours, and the model outputs recorded every hour. The observed evolution of greenhouse gases $CO_2$, $CH_4$ and $N_2O$ were
145 incorporated as recommended in Jerez et al. (2018), varying $CO_2$ between 353 and 390 over the simulated period.

## 2.2 Methods

This contribution focuses on the impacts of ARI and ACI on precipitation. Hence, the climatologies for precipitation amount, number of days with precipitation over a given threshold and cloudiness of the different experiments were intercompared for the BASE, ARI and ARCI simulations. The ERA5 (Hrarsbach and Dee, 2016) reanalysis data was used to calculate the added
value of the aerosol experiments, since it has already been validated for precipitation (Albergel et al., 2018; Christensen et al., 2019; Hwang et al., 2019). Also represented are the comparison of the annual and seasonal climatologies for other atmospheric fields such as sea level pressure (slp), geopotential height (Z) and temperature (T) at 1000,750 and 500mb, maximum minimum temperatures (tasmax,tasmin), daily temperature range (dtr) and solar radiation at surface (rsds) as well as mean temporal fields of the particulate matter (PM10,PM2.5), BC (black Carbon) and AOD. All these fields as presented as supplementary material.

The statistical significance of the differences between the climatologies reproduced by the simulations was checked using a Bootstrap method with 1,000 repetitions and applying a p-value < 0.05. Further details of this method can be found in Milelli et al. (2010).

In order to assess the relationship between the obtained changes in precipitation and different variables representing the aerosol load PM10 (Particulate Matter <10$\mu$m), PM2.5 (Particulate Matter <2.5$\mu$m), AOD at 550nm, the ratio between PM2.5 and PM10 (hereinafter called PMratio), several events (days) are grouped according to their intensity and extension. The intensity of an event is defined as the minimum value given by a threshold variable that the simulation cells must meet; the extension of an event is defined as the number of cells meeting the previous condition.

The relative differences (ARCI-BASE)/BASEx100 between the experiments are shown on a two-dimensional heat map, where the axes denote extent and intensity. The number of days on which the criteria defined above are met is indicated inside each element of the matrix. The total number of days analyzed is 7305, corresponding to the 20 years simulated. This type of graph allows us to identify whether there is a relationship between the different variables and the magnitude of the change, and to establish the relative importance of each factor involved. In the intervals where a relationship appears, a multiple linear regression fit was made, giving the multiple correlation coefficient as indicator of the skill of the relationship.

On the other hand, the effect of aerosols could depend on the area and affect weak and strong precipitation events differently (Rosenfeld et al., 2008). The series of relative differences between the ARCI-BASE simulations were generated for common and non-common days with rainfall exceeding a certain threshold for all points in the domain. The threshold ranges from 0 to 20mm/day on a non-linear scale (with a higher density of values near 0) with a total of 41 values. In order to investigate areas where the effect of aerosols on precipitation could be different, a clustering method was applied to the constructed series. The algorithm used for the spatial classification is similar to that used in other works (Jiménez et al., 2008; Lorente-Plazas et al., 2015) and is composed of several steps. First, an analysis of the principal components (Von Storch, 1999) is made and applied to the correlation matrix of the constructed series. Second, a two-step clustering method is applied to a number of the retained principal components. A hierarchical method is applied as a previous step; in this case, Ward's algorithm (Ward Jr, 1963). This classification provides the number of clusters and initial seeds (also called centroids) for the final step, application of the non-hierarchical method K-means which optimizes the grouping (Hartigan and Wong, 1979). Further details about the algorithm can be found in Lorente-Plazas et al. (2015). Finally, the mean regional series are calculated as the average of series belonging to a cluster (which corresponds to a spatial region in this study).

## 3 Results and discussion

### 3.1 Precipitation differences in ARI and ARCI simulations

The sensitivity of precipitation to the aerosol treatment in climate simulations is analyzed by comparing BASE, ARI and ARCI simulations over Europe over a 20-year period. The differences between ARCI-BASE (ARI-BASE) in spatially-averaged total precipitation are limited, around 0.5% (0.1%). Figure 2 shows the relative differences with respect to BASE in the mean annual rainfall. The results depict a large spatial variability with differences ranging from 10% to -10%. Two zones with opposite

behaviors are identified: (1) the central and eastern part of Europe, with a precipitation decrease of up to 8% (statistically significant, p<0.05), and (2) the eastern Mediterranean area, with increases of up to 10% (although the changes are not significant, p> 0.05). Other areas, such as the Iberian Peninsula, present a strong spatial variability (e.g. rainfall increasing over the Mediterranean coast and decreasing over northeastern areas). Overall, the fact of introducing ARI and ACI interactions leads to a redistribution of the annual precipitation. The most remarkable difference is a reduction of annual precipitation over central Europe for ARI which is enhanced when the more intense and spatially extended ACI interactions are included. This reduction of precipitation is linked mainly to a reduction of the number of days with precipitation $> 0.1$mm ($N_{p01}$) and clouds at low level (CLL); indeed, the most significant and widespread changes are obtained for CLL. Moreover, a statistically significant increase of $N_{p01}$ appears over the eastern Mediterranean, but in this case only in ARCI experiments linked to an increase of CLL. At seasonal scale (see Supplementary Material, Figures S6-S11 for further information), the decrease of precipitation, CLL and $N_{p01}$ in central Europe is reproduced in all seasons but summer. In addition, the increase in the eastern Mediterranean is reproduced throughout the year, with the largest absolute changes in winter.

These changes are also related to others in several variables: for instance, *rsds* decreases in ARI and ARCI experiments mainly over the southern half of the domain, due to the higher AOD. However, there are some parts of central Europe where *rsds* rises due to the decrease of clouds, especially in autumn and spring (Figure S12,S13). Changes in temperature are different for *tasmax* and *tasmin* (Figures S14 and S15). They are larger for *tasmax*, especially in ARCI, reaching differences around 0.5K and presenting quite similar spatial patterns to those of CLL, while *tasmin* do not present any correlation with CLL. The most remarkable changes are obtained for *dtr* with a pattern characterized by an important increase in the north (lower CLL) and a decrease in the south (higher AOD) (Figure S16). The modification of energy fluxes also affects circulation. The *SLP* fields, as well as *Z* at several levels, also show statistically-significant sensitivity to ARI and ACI effects (Figures S17, S18,S19). Here, the most remarkable features are the large differences between the ARI and ARCI experiments. ARCI shows a noticeable increase of *slp* in the central and northern parts of the domain with respect to ACI. This behavior is also appreciated for *Z*. Finally, it is worth highlighting that ARI and ARCI also indicate a rise in temperature over northern and central Europe. This might imply that simulated changes in precipitation can also be indirectly affected by changes in atmospheric circulation. This fact could make it more difficult to establish the relationship between changes in precipitation and changes in the treatment of aerosols in our experiments.

In order to investigate the variations in the regimes of precipitation, the changes in the number of rainy days are estimated. Figure 2 (and Figure S9) shows the relative differences in the number of days with precipitation $> 0.1$mm. The patterns of differences are similar to those of averaged precipitation, implying that the reduction in precipitation is mainly caused by the decrease in the number of rainy days. However, there are some noticeable exceptions. The relationships in the two large areas mentioned above are direct; that is, higher rainfall is linked to a larger number of precipitation episodes. However, there are areas where the relationship is inverse, a higher(lower) number of days implies less(more) precipitation. Analysis of the low clouds in the domain (Figure 2 and S10-S11) shows a pattern similar to the aforementioned ones. This might indicate that the effects of both ARI and ACI can play very different roles in cloud properties and therefore in precipitation, depending on the target area. This issue will be addressed later in this contribution.

## 3.2 Evaluation against ERA5 reanalysis

The added value of incorporating on-line aerosol interactions and complex aerosol physics into the model was calculated by
analyzing the differences in precipitation, number of rainy days and low clouds between the simulations and the reanalysis of
the European center ERA5 (Figure 4). Overall, WRF-Chem (in both the BASE and ARCI simulations) tends to underestimate
precipitation over the European Mediterranean region and along the coasts of the Nordic countries, while it overestimates
rainfall in the rest of the domain. These patterns are analogous for all the variables analyzed. Looking only at the areas where
the differences are significant, the ARCI simulations slightly reduce the differences in the spatial distribution. However, the
differences between ERA5 and ARCI are much larger than the differences between ARCI and BASE .

Despite this, as previously noted (Figure 2), the ARCI experiment introduces significant differences with respect to the
BASE simulation over central Europe. These differences reach values of about 5% in the number of rainy days. A relationship
between aerosols in these areas and the aforementioned changes might therefore be expected in spite of the induced changes
in the dynamics. This relationship is explored in the following section of this contribution.

## 3.3 Relationship between aerosol physical properties and precipitation

In order to understand the contribution to changes in precipitation of the different types of aerosol, the differences in rainfall
were assessed by choosing a set of episodes. These were selected according to the value of variables representative of the size
and concentration (PM10 and PM2.5), ratio (PMratio) and impacts on radiation (AOD) of the aerosols, as well as the spatial
extension of the event.

Figure 5 shows the relative changes for the different sets of episodes for AOD at 550nm (AOD550) (b), PM10 (d), PM2.5 (c)
and the PMratio (d). The calculations were conducted using only the points with significant differences (Figure 2). Figure 5a
shows the relative changes (ARCI-BASE) in the number of rainy days for different sets of episodes, selected by choosing the
extension/size of the episode (number of grid points) of the cells exceeding a value of PMratio (values from 0.2 to 0.8). In a
range of intensities, quasi-linear relationships appear. Figures 5b-e show these relationships for the different variables.

The lower left box of Figure 5e indicates that 5970 out of 7303 days present a PMratio > 0.64 (y axis) achieved in more than
180 cells of the domain (x axis). When calculating the differences in ARCI-BASE precipitation in the 5970 days accomplishing
that condition (PMratio > 0.64 in more than 180 cells of the domain), the differences in rainy days over those cells is around
4%. Thus, e.g., the number of days in which PMratio is > 0.75 in more than 280 points is 1030 and the reduction in the number
of rainy days is 8%. Following with PMratio (Figure 5e), the higher the intensity, the larger the reduction in the number of rainy
days; and the greater the extent/size of the event, the larger the reduction in rainy days (e.g. reaching the maximum reduction
around 15%) . Indeed, the multiple regression coefficient between the different variables is R = 0.80.

For AOD550 (Figure 5b), the results show that higher AOD550 values lead to a lower reduction in the number of rainy
days. The changes are small (under 2%) although the relationship is clear (R = 0.78). The results are analogous for PM2.5
(Figure 5c) but the relationship is less clear (R = 0.53). For PM10, the changes are higher but with a less clear relationship

(R = 0.40). However, the relationships with the PMratio (Figure 5e) are important and significant (R = 0.80). Therefore, an important conclusion is that the variable with the largest impact on the number of rainy days is the PMratio in that area.

The possible physical explanation for this behavior in this area is that the higher the PMratio (Figure 3), the higher the concentration of small particles changing the properties of the clouds (mainly low clouds) (Figure 2; a reduction of low cloudiness over central Europe) and leading to a clearer atmosphere. This results in higher temperatures and an increase in the condensation level, leading to a reduction in the number of rainy days and therefore a decrease in the amount of precipitation (direct and semidirect effects). As noted in Figure 2, the reduction of CLL also occurs in the ARI experiment. This could be explained by the atmospheric warming caused by the radiation absorption of dark atmospheric aerosols (black carbon), causing the above effect. The stronger signal in ARCI can be attributed to the addition of both processes. On the other hand, a high concentration episode of PM2.5 can occur together with a PM10 event, decreasing the PMratio. Therefore, the better relationship with PMratio could be related to coarse aerosols enhancing precipitation, and thereby opposing the effect of smaller aerosols.

## 3.4   Regional role of aerosols on precipitation

As noted previously, the relationships between changes in precipitation, number of rainy days and cloudiness, are different in different regions of our domain. Therefore, the role of aerosols, analyzed considering either their nature or their concentration, causes different changes in precipitation regimes. In order to quantify this effect, the series of relative changes in the number of rainy days were constructed at each point for different thresholds ranging from 0.1 to 20mm/day. The grouping method described in the methodology section was applied to this series, obtaining 5 different regions (Figure 6). The clusters are listed according to the number of grid cells of each group, with Cluster 1 the most numerous and also the most dispersed. The centroid series (average series of regions) are represented in Figure 7. The (green) filled circles indicate that the relative differences between the ARCI and BASE experiments are significant.

Cluster 1 does not present a clear pattern, covering most of the points of the Atlantic Ocean and southern Europe. This area has very low, non-significant differences, with values between 0.5% and -2.5%. Therefore, the effect of including aerosol-cloud interactions in this area does not practically affect precipitation. Clusters 2 and 5 have a similar behavior. In both zones, there is a decrease in precipitation for almost all thresholds except the most extreme rainfall events where precipitation increases. In Cluster 2, the changes range from -2% to -4%, with significant differences for low thresholds (up to 2mm/day). In the case of Cluster 5, the differences are always significant and much larger. The maximum reduction is obtained for episodes of precipitation above 14mm/day, reaching relative changes in the precipitation of the entire area of around 12%. Note that Cluster 5 almost coincides with the area previously analyzed (significant differences, Figure 2).

Clusters 3 and 4 have a different behavior. In these regions, an increase in precipitation occurs when including ARCI. Cluster 3 does not have a clear spatial pattern, with points scattered along the entire domain. For low thresholds, there are no significant changes; for high thresholds, there is a very significant increase in precipitation with significant relative changes (e.g. 5% for a threshold of 8mm/day). For higher thresholds, the relative changes are close to 20%. However, this result should be analyzed with caution because of the lack of spatial structure, although from the statistical point of view there is a coherent increase in

moderate and intense precipitation events that can be explained by some physical processes presented in the literature (Khain et al., 2008).

Finally, Cluster 4 shows a clear spatial pattern, with most of the points concentrated in the eastern Mediterranean. Over this area, the range of thresholds between 1 mm/day and 5 mm/day presents significant differences; while for thresholds > 5mm/day, the series remain constant around 4.5% and the statistical significance disappears.

Therefore, the role of the aerosols in precipitations shows a clear spatial dependence, affecting strong and weak precipitation differently. Over Regions 2 and 5, which cover northern, central and eastern Europe, ARI and ACI interactions tend to reduce precipitation. This reduction is significant for almost all events below 15mm/day. In the Mediterranean area, and especially in the eastern Mediterranean, rainfall increases in the ARCI experiment, mainly due to the increase in the number of days with rainfall below 5mm/day. Meanwhile, in Cluster 3, the total rainfall undergoes very variable changes, but fundamentally an increase in moderate and strong rainfall events.

## 3.5   ARI vs. ARCI relevance for modifying precipitation

In order to better understand the processes involved in each of the areas, the absolute annual values and differences between ARCI and ARI are analyzed in terms of the concentrations of PM10, PM2.5 and PMratio (Figure 3). This will allow us to discriminate which processes (aerosol-radiation or aerosol-cloud interactions) are most relevant. As commented above, Figure 2 shows the differences in ARCI-BASE, ARI-BASE and ARCI-ARI analyzing precipitation (number of days exceeding 0.1 mm/day and total amount) and the cloud cover at low level. In the case of Cluster 5, both simulations provide a reduction in the number of days of precipitation. Therefore, both ARI and ACI affect precipitation in the same direction. ARI causes an increase of temperature at low levels (see temperature at 850 hPa, Figure S21, in the Supplementary Material), especially in autumn and spring, leading to a reduction in clouds and precipitation. A possible explanation would be that the ARCI experiment enhances this effect by the higher concentration of small particles modifying the properties of the clouds, inhibiting precipitation processes again by reducing clouds due to microphysical processes, since over this area there is a prevalence of small aerosols (see PMratio in Figure 3 and Figures S1-S5 of Supplementary Material). Another possible explanation could be linked to the changes in circulation which reduce both cloudiness and precipitation (see Supplementary Material, Figures S17-S22).

Finally, the increase in precipitation and cloudiness in Cluster 4 could be associated with larger values of PM10 (large condensation nuclei). In this case, the ARI effects are almost negligible during the year. However, the ARCI experiment shows a clear positive difference with respect to the BASE case and ARI. Figure 8 shows the relative difference in the concentration of PM10 between ARCI and ARI, and the differences in the number of rainy days with precipitation > 1mm/day. The points are distributed in a quasi-random way with respect to 0. The cells of the whole of Cluster 4 show a bias towards positive values for changes in precipitation and a decrease for PM10. Focusing only on the eastern Mediterranean of cluster 4 (yellow points), the relationship is clear. Most of the points showing an increase in precipitation undergo a decrease in PM10. A plausible explanation is that in these areas the PM10 load is high due to the intrusion of desert dust and sea-salt aerosols. The difference between the ARCI and ARI simulations is the activation of the aerosol-cloud interaction mechanism, using the

aerosols calculated online as CCN to form clouds, while in ARI, the CCN are prescribed at a fixed value. The PM10 used to form clouds in ARCI will no longer be counted in PM10 because of in-cloud scavenging. Therefore, a decrease in PM10 occurs and this coincides with an increase in cloudiness. In addition, the increase of precipitation will also decrease PM10 due to wet deposition. Note that the patterns do not completely coincide, with the precipitation pattern shifted slightly to the north (see the comparison in Figures 2). This can be attributed to the displacement of the cloud masses in such an area. This behaviour can be attributed to the role of giant aerosol particles in warm rain initiation (Johnson, 1982), increasing stratiform precipitation by dust through deposition growth (Gong et al., 2010) or enhanced drizzle formation in stratocumulus (Feingold et al., 1999).

## 4  Conclusions

The effect of atmospheric aerosols on regional climate simulations nowadays presents many uncertainties due to complex and non-linear processes which depend on a wide variety of factors. The quantity, size and optical properties of aerosols condition the modification of the radiative budget and, therefore, many other derived variables such as local temperature, cloudiness or precipitation. In addition, the amount of available moisture determines the size of the water droplets based on the amount and type of aerosols available. Atmospheric aerosols also affect the size and optical properties of the clouds, which also modify the radiative budget. Moreover, these processes can spatially redistribute the precipitation regimes, allowing rainfall in different areas or provoking changes in its intensity. Despite the importance of the problem from a climatological point of view, there is a scarcity of scientific contributions that have studied these issues. The large increase in the computational time needed to include ACI and ARI interactions in regional climate simulations has traditionally hampered the works covering this analysis from a climatic perspective.

In order to address the aforementioned issues, a set of regional climate simulations were conducted for the period 1991-2010 without on-line aerosol-atmosphere interactions (BASE), with ARI and with ARI+ACI (ARCI) parameterizations in an on-line coupled model. All the simulations cover the domain of Europe defined by the Euro-CORDEX initiative. This analysis focused on average precipitation, number of precipitation days over a certain threshold and cloudiness. In addition, the effects on other variables such as temperature at different levels, geopotential height, radiation at surface, and sea level pressure are presented as supplementary material (SM).

When introducing the ACI and ARI interactions, the spatial average of the total rainfall does not differ from the BASE scenario. However, there is a spatial redistribution of such precipitation. Although there are changes in several places throughout the domain, the largest modification occurs in central Europe, where a decrease in precipitation is found as a result of activating the aerosol-radiation and aerosol-cloud interactions. The behavior is the opposite in the eastern Mediterranean, where the effects of aerosol-cloud interactions prevail. These results are reproduced by analyzing the number of days of precipitation > 0.1mm, with very similar patterns. However, there are areas where the relationship between precipitation and number of rainy days is not straightforward.

When the results are compared with ERA5, the BASE simulation tends to overestimate rainfall across the domain except in some areas of Mediterranean and Nordic countries. When the ACI interactions are incorporated into the modeling setup, these differences are reduced, although quantitatively this improvement is limited.

The results obtained for the number of precipitation days $> 0.1$mm were related with different aerosol variables (AOD550, PM2.5, PM10 and PMratio). That relationship shows a highly non-linear behaviour, although a regime where the linear approximation is acceptable was also identified. For central Europe, in the linear regime, the intensity and extension (size) of the PMratio events have a direct relationship with the increase of the differences in the number of rainy days.

Although the previous conclusion is limited to the number of days of precipitation $> 0.1$mm, it becomes interesting to check the relationship for other thresholds. Five types of behavior throughout the target domain were identified by analyzing several precipitation thresholds. Aerosols contribute positively or negatively to precipitation depending on the area and the intensity of precipitation. The available humidity, efficiency of the CCN and type of aerosol (size, optical properties, shape) are the most important factors conditioning the type of behavior. In the experiments conducted, the inclusion of ARCI leads to a reduction of precipitation in all regimes in northern-central and eastern parts of Europe. However, in the eastern Mediterranean, precipitation increases due to the increase of days with rainfall $< 5$mm/day. Positive changes for moderate and strong rainfall regimes are also found over some areas (Cluster 3, which is a very dispersed area). Although this finding can be identified with the so-called *deepening effect* (Stevens and Feingold, 2009), relating aerosols to an increase of precipitation for some convective events, this statement should be considered with caution because of the lack of spatial structure of this cluster. The rest of the areas are barely affected.

Some of the changes obtained can be related to the direct, semidirect and indirect effects of aerosols on clouds. The reduction of precipitation over some areas could be linked to both atmosphere warming and excess of CCN. The radiative processes have the ability to change the thermodynamic environment due to the absorption of radiation by fine dark particles (mainly black carbon), stabilizing the environment or increasing the condensation level. The excess of CCN leads to small drops producing a depletion in precipitation . In principle, this would increase the lifetime effect; however, the experiments presented here show an extra depletion of cloudiness, possibly related to faster evaporation of the water drops. All these processes are associated with a high concentration of fine aerosols with respect to coarse particles. On the other hand, the effects of coarse aerosols (PM10, giant condensation nuclei) seem to be the complete opposite. These particles seem to enhance precipitation processes, especially increasing light precipitation events (Feingold et al., 1999) or accelerating precipitation development. Sometimes both processes (semidirect and indirect) overlap, with a negligible net effect.

In conclusion, the effect of aerosols on climatic variables is varied and complex and further studies on this topic are needed in order to (1) reduce the uncertainty associated with the inclusion of aerosols in regional climate experiments; and (2) better understand the physical and microphysical processes leading to changes in precipitation. This contribution demonstrates from a modeling approach that changes in the concentration, extension and type of aerosols alter the precipitation regimes and amount in different ways. These changes are spatial- and seasonal-dependent and in agreement with other works (e.g. Li et al. (2019)). The inclusion in regional climate experiments of on-line aerosols, as well as cloud-aerosol interactions, alter precipitation patterns as well as other surface and upper air variables (Pavlidis et al., 2020; Jerez et al., 2020b) and could differ from other

approximations such as using AOD climatologies or prescribed CCN (Nabat et al., 2015). It would be interesting to see to

390 which extent other regional models would reproduce the current results for the Euro-CORDEX region in order to analyze the possible model dependence of the results. Future research aimed at disentangling the effects of aerosols on regional climate simulations should be devoted to understanding the role of regional and large scale circulation (regimes), possible feedbacks and overlapping processes.

*Author contributions.* JML-R, JPM and PJ-G designed the research; JML-R performed the experiments; JML-R, JPM, SJ and RL-P analyzed

the outputs from experiments; LP-P contributed to the design of the numerical experiments; and JMLR and JPM wrote the paper, with inputs from all coauthors.

*Competing interests.* The authors declare that they have no conflict of interest.

*Acknowledgements.* The authors thank the WRF-Chem development community and the G-MAR research group at the University of Murcia for the fruitful scientific discussions.

*Financial support.* This research was funded by the European Regional Development Fund-Fondo Europeo de Desarrollo Regional (ERDF-FEDER), Spanish Ministry of Economy and Competitivity/Agencia Estatal de Investigación grant number CGL2017-87921-R (ACEX project), Spanish Ministry of Science, Innovation and Universities grant number RTI2018-100870-A-I00 (EASE project) and CLIMAX project (20642/JLI/18) funded by the Seneca Foundation-Agency for Science and Technology in the Region of Murcia. LP-P thanks the FPU14/05505 scholarship and the Spanish Ministry of Education, Culture and Sports. JML-R acknowledges the FPI-BES-2015-074062

grant from the Spanish Ministry of Science.

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

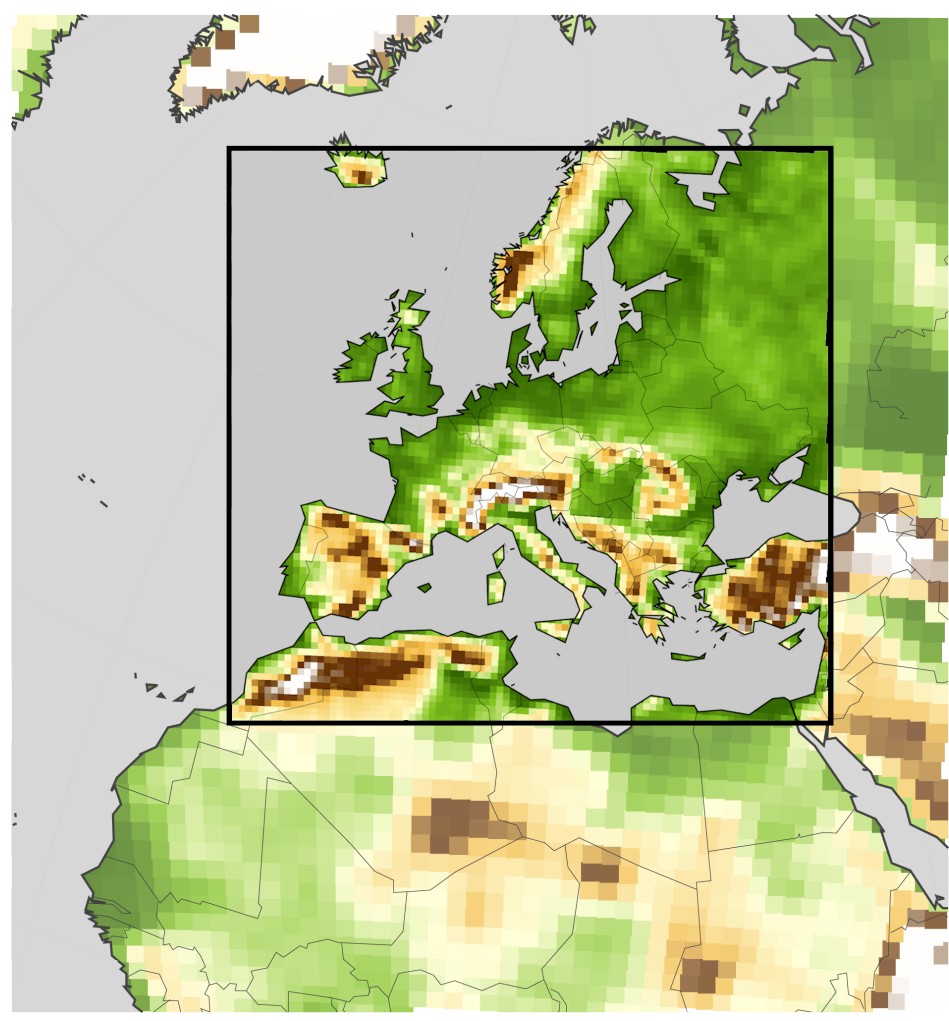

**Figure 1.** Simulation domains covered in the experiments. The inner Euro-CORDEX domain is boxed in the Figure.

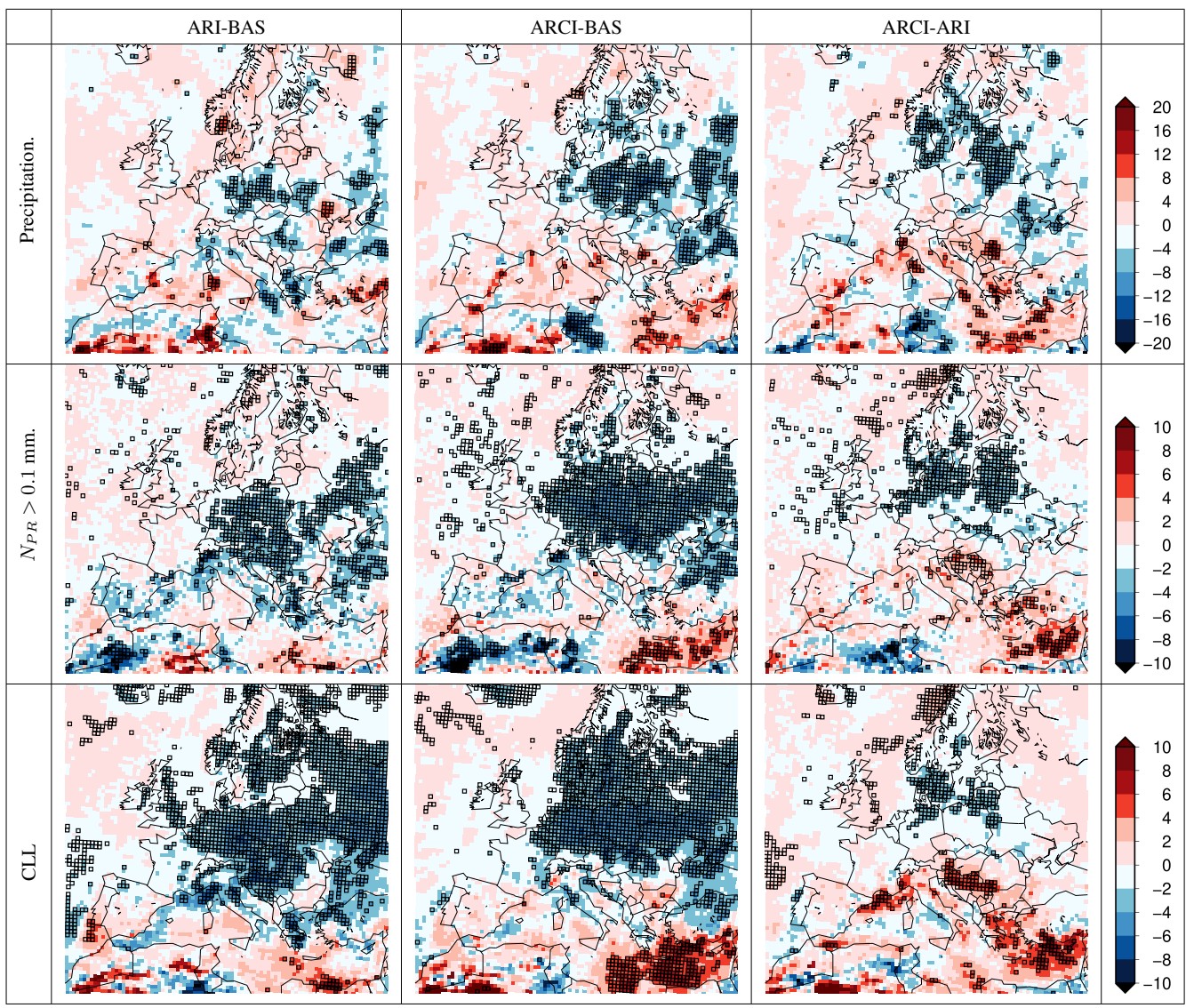

**Figure 2.** Relative differences for precipitation between ARI and BASE (first column), ARCI and BASE (second column) and ARCI and ARI (third column), total precipitation (first row) number of days of precipitation > 0.1mm (second row) and low clouds (Third row). Squares indicate points whose differences are significant for a p-value of 0.05. The analysis was conducted for the mean values of the period 1991-2010.

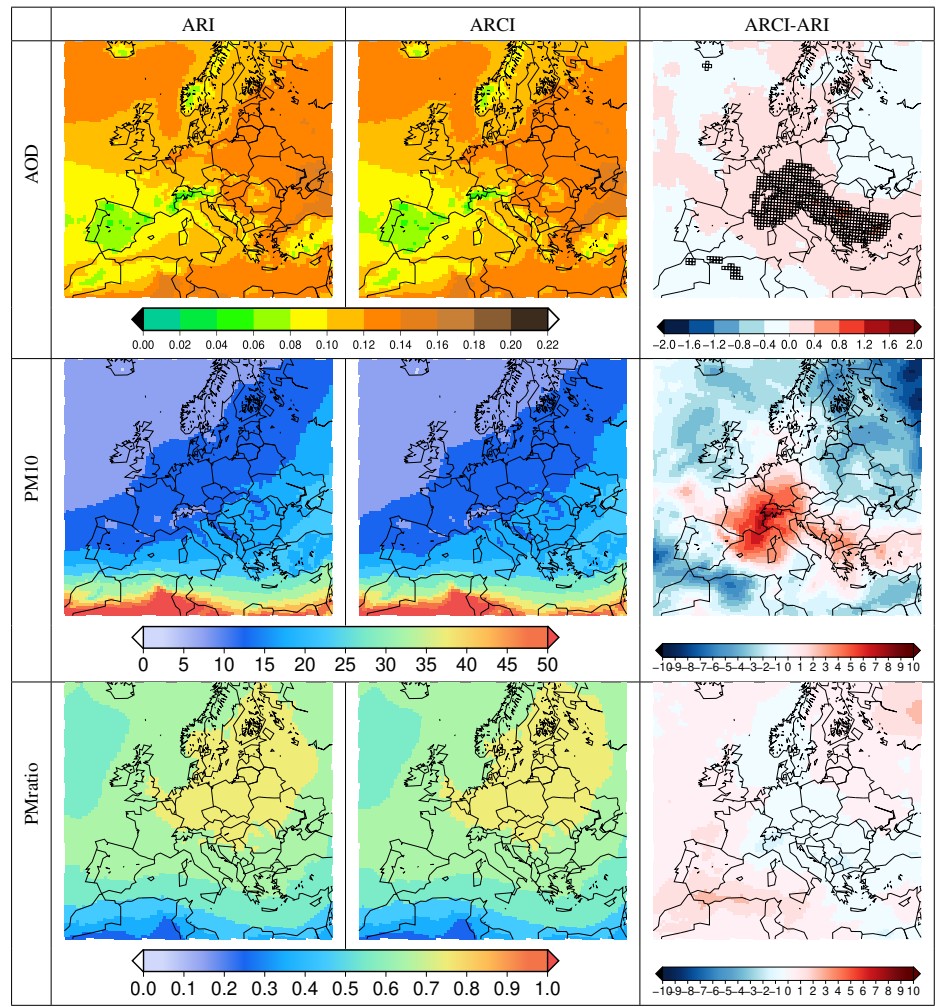

**Figure 3.** AOD, PM10 ($\mu g/m^3$) and PMratio mean annual values for ARI and ARCI and their differences (%).

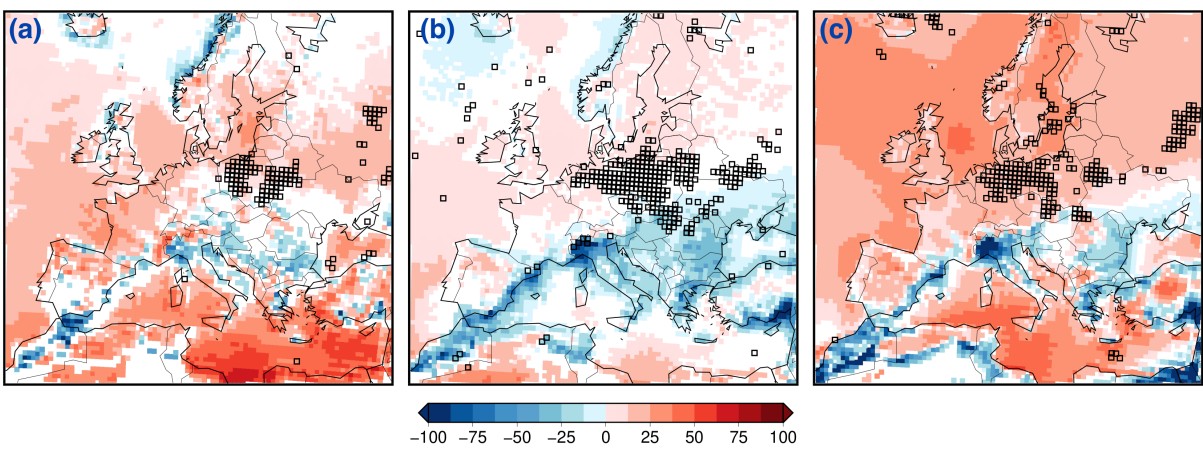

**Figure 4.** Significant relative differences (colors) between ARCI and ERA5 for (a) precipitation, (b) number of days of precipitation > 0.1mm and (c) clouds at low levels. ). Squares indicate statistical significant differences (p < 0.05). The analysis was conducted for the mean values of the period 1991-2010.

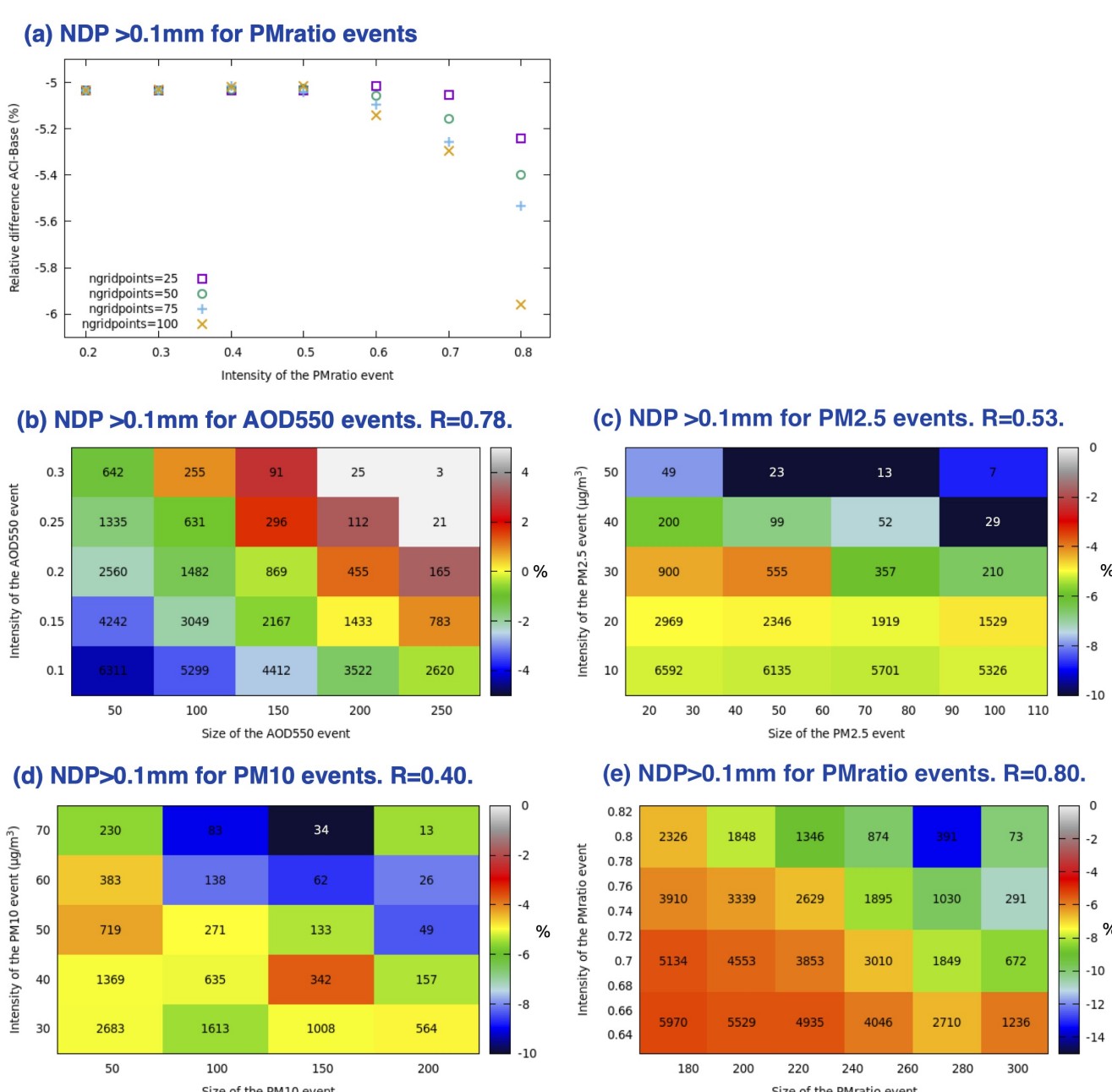

**Figure 5.** Relative difference (colors) in the ARCI–BASE simulations for the 1991-2010 period based on (b) the intensity and size of AOD550 events, (c) the intensity and size of PM2.5 events, (d) for events of PM10 and (e) for those of PMratio. The calculation is made for the domain cells with significant ARCI-BASE differences for the number of days with precipitation > 0.1mm (Figure 2b) and only for the zone where the non-linear behavior begins (>0.6) in Figure 5a (id. to the other variables). The number inside the boxes indicates the number of days meeting the corresponding criteria of intensity and extent of events. R denotes the multiple regression coefficient resulting from a multi-linear adjustment of those values.

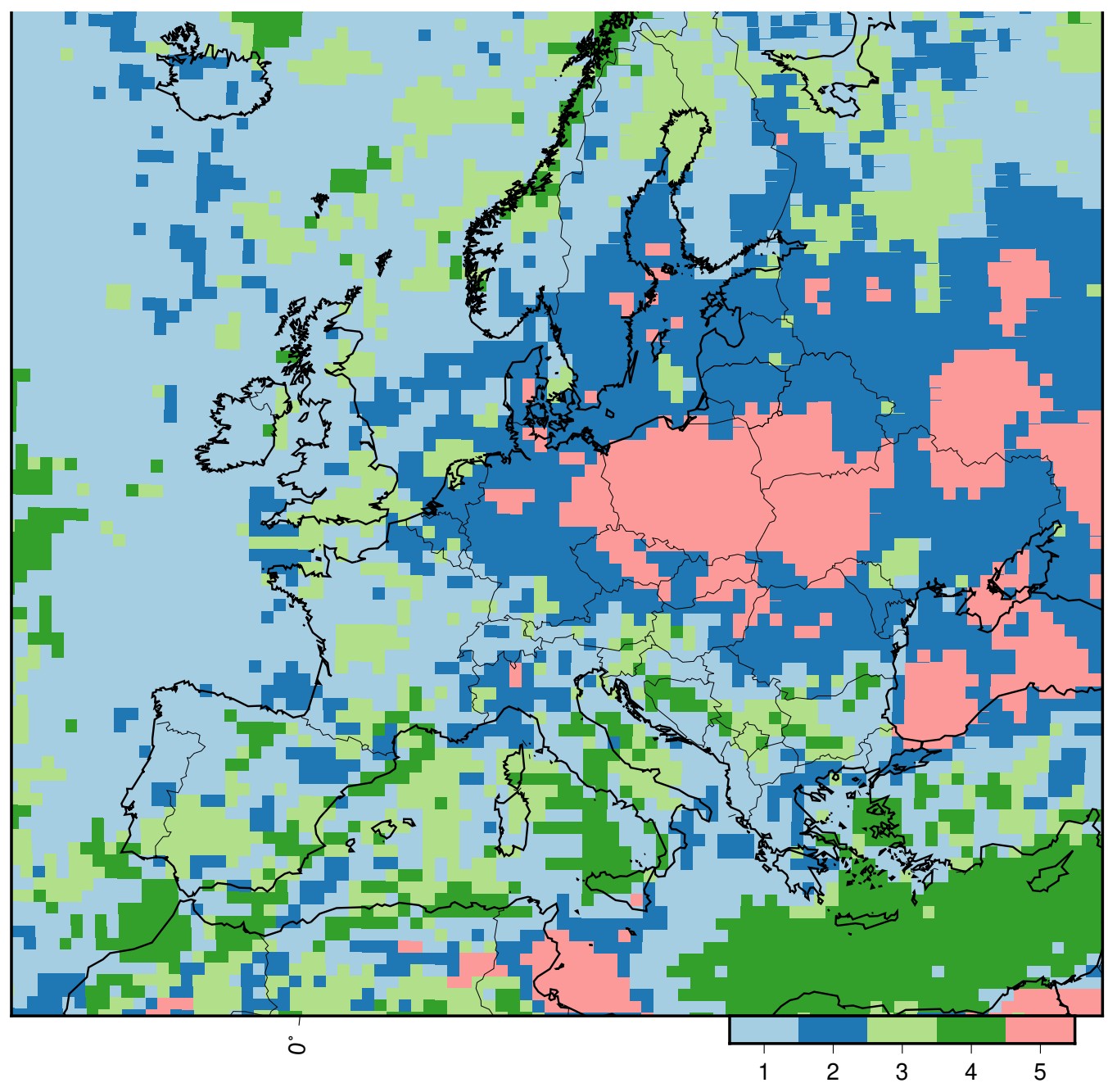

**Figure 6.** Cluster analysis of rainy days: each color depicts a cluster with different behaviour of the ARCI-BASE difference in number of days of precipitation over a threshold running from 0.1mm to 20mm/day for the period 1991-2010.

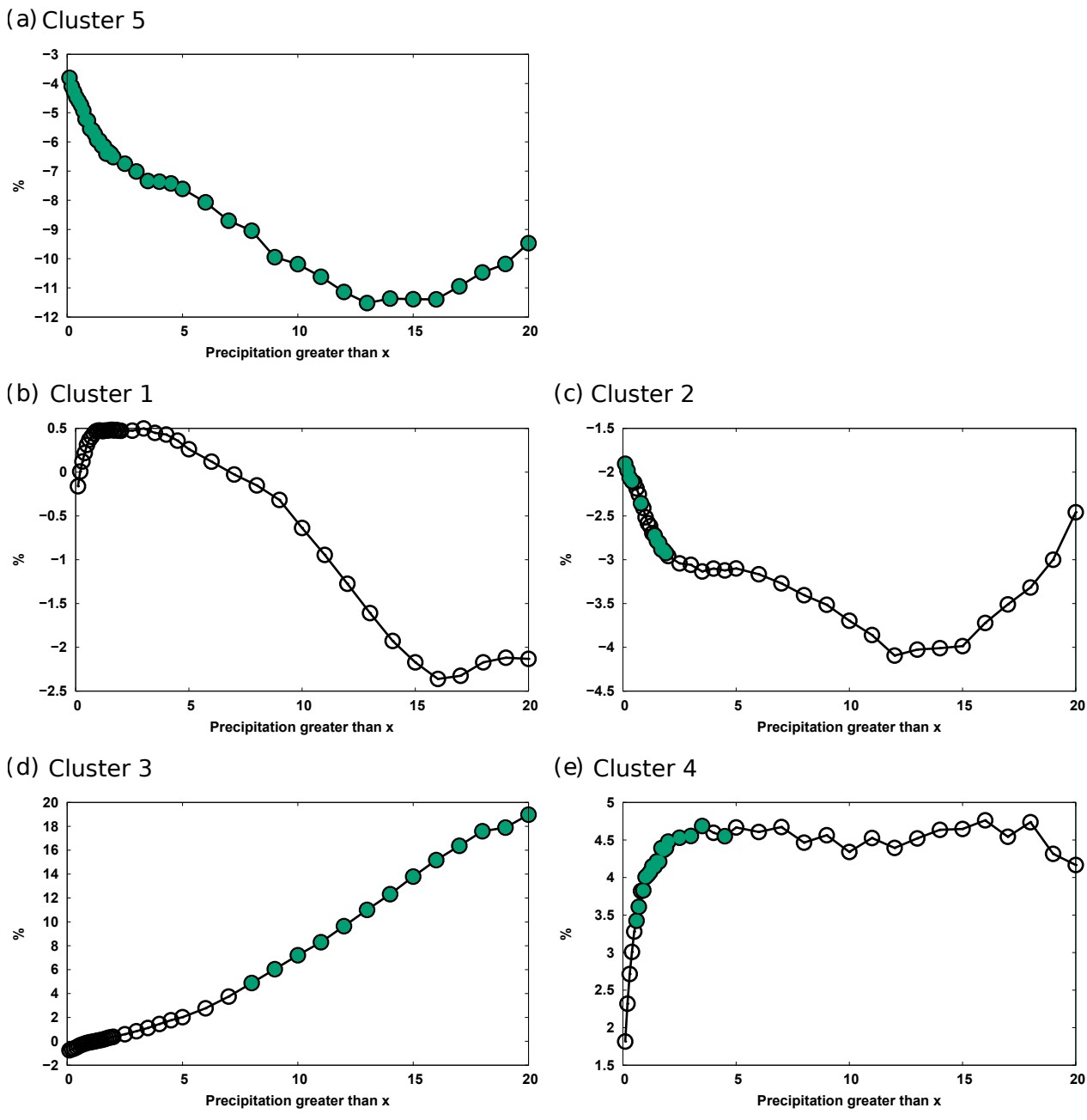

**Figure 7.** Series of relative differences between ARCI and BASE based on different thresholds in rainy days for the different regions (Figure 6). Green circles denote the thresholds for which the differences are significant (p-value $< 0.05$).

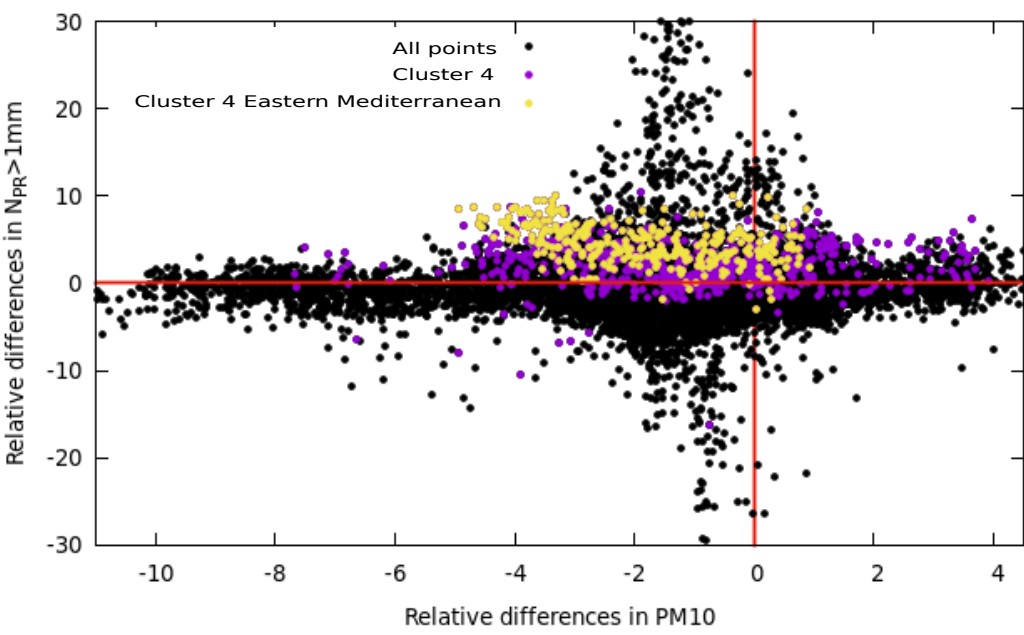

**Figure 8.** Relative differences (ARCI-BASE) of the number of days of precipitation $> 1mm$ versus PM10 (ARCI) for all the cells of the domain (black), for Cluster 4 (violet) and Cluster 4 but only in the Mediterranean (yellow).