# Peer review of "Precipitation response to aerosol-radiation and aerosol-cloud interactions in regional climate simulations over Europe"

_Atmospheric Chemistry and Physics, 2020_

## Referee Comment (RC1) · Anonymous Referee #1 · 7 Jun 2020

**General comments**

This paper documents relatively long (20 years) simulations of precipitation with the regional WRF-Chem model driven by ERA20C reanalysis. Two simulations with interactive aerosols (one including only the aerosol direct radiative effect, another also the effect on cloud microphysics) are compared with a baseline simulation with fixed aerosols. It is found that the use of interactive aerosols decreases precipitation in Central/Eastern Europe and increases it in the Eastern Mediterranean. Detailed analyses regarding the number of days with precipitation with different thresholds are carried out.

[Figure]

The treatment of aerosols in regional climate models is often rather primitive, and therefore I think the authors have carried out a valuable set of experiments. At the same time, I cannot recommend the publication of this paper in ACP, unless substantial improvements are made in the analysis and reporting of the results. The reasons for my concerns are outlined below.

**Major comments**

1. My primary concern regarding this paper is that while it documents in some detail how precipitation changes, the physical interpretation of the results is rather lacking. The paper fails to properly address the question, what are the physical mechanisms leading to these changes in precipitation. Only a few cursory statements are made in this respect. The changes in precipitation could be caused by several mechanisms. They could arise through the impact of aerosols on cloud microphysics, or their impact on surface temperature (which could suppress convection), or through changes in large-scale meterology (although the latter are probably small due to the use of nudging in the outer model domain).

2. To make it easier for the reader to interpret the findings, simulation results should be shown for additional physical quantities. It is very difficult to understand precipitation by looking at precipitation (and low clouds) alone. Most obviously, the paper should start with briefly showing how the aerosol fields (AOD, CCN, and aerosol radiative forcing, if available) differ from the baseline simulation, since these differences are the root cause for the changes in precipitation. I realize that some of this information is probably available in the cited papers by Palacios-Pena et al., but this paper should be able to stand alone — it should not be the reader's task to hunt for necessary information in other papers. Furthermore, changes in surface temperature are potentially important for convection, and they are referred to at a couple of occasions, but it would be

better to actually show them. Other quantities that should be checked (and possibly shown, if their changes seem important for precipitation) include meteorological fields like surface pressure, relative humidity and mid-troposphere vertical vecolity.

3. The interpretation of the results is also complicated by the fact that data for all seasons are lumped together. Yet, the processes generating precipitation, and potentially their sensitivity to aerosols, depend strongly on the season. Especially concerning central-eastern Europe, which shows the clearest signal in precipitation, convective precipitation dominates in summer, while stratiform precipitation associated with synoptic weather systems dominates in winter. I recommend that the authors first look at precipitation on a season-to-season basis (at least distuinguishing between the warm and cold seasons), and then focus the detailed analysis on the season(s) with the most meaningful signals.

4. While the authors have conducted both ARI and ACI simulations, the ARI results are not discussed much, except for Fig. 6. I strongly recommend to show the ARI−BASE and ACI−ARI differences, at least for the time-average precipitation in Fig. 2. It is vital information for understanding to which extent the precipitation changes arise from aerosol direct and indirect effects.

5. There are rather many issues with the use of English language. At the end of the review, I list cases which I found disturbing for correct understanding of the text. This is not intended to serve as a complete language check.

**Detailed comments**

1. line 16: should this be "eastern Mediterranean"?

2. line 29: Can you add a reference to a publication listing the WCRP five major scientific challenges?

3. line 32: I suggest replacing "The main tool" with "One of the main tools". The IPCC AR5 estimates of aerosol radiative forcing use satellite observations to adjust model-based results.

4. lines 40–44: A key point of the convective invigoration mechanism of Rosenfeld et al. (2008) is that the slower cloud-droplet-to-rain conversion allows the droplets to be transported above the freezing level, and therefore, the latent heat released in freezing makes the convection more intense.

5. lines 46–49: It would be useful to give a bit more information on the cited studies (e.g., which regions were considered?).

6. lines 59–60: "and abundant number of cloud condensation nuclei (CCN) (Forkel et al., 2015) high enough for clouds to form without this variable being a limited factor". In fact, the lack of CCN is almost never a limiting factor for cloud formation (this could perhaps happen in remote marine locations in very specific conditions). However, a low CCN value may result in clouds that precipitate more readily, which can reduce the cloud lifetime and therefore the average cloud fraction.

7. line 67: "black anthropogenic aerosols". Do you mean black carbon, or absorbing anthropogenic aerosols in more general? Furthermore, this paragraph gives the impression that anthropogenic aerosols cause warming and natural aerosols cause cooling, which is misleading. Many anthropogenic aerosols, most prominently sulfates, are largely non-absorbing, so the total effect of anthropogenic aerosols is probably one of radiative cooling.

8. lines 112, 116: You mention the use of both the Goddard shortwave radiation scheme and the RRTMG scheme. To my knowledge, these are different radiation schemes. Please explain.

9. lines 127-129: While AOD (it should be "aerosol optical deth") has been evaluated by Palacios-Pena et al. (2020), it would be definitely good to show the time-mean AOD fields also in this paper (see major comment 1).

10. line 163: correlation matrix of what?

11. lines 174-179: The spatial redistribution of precipitation is interesting, but is very difficult to figure out why it is happening, based on the information given in this paper. Please see the major comments 1-3.

12. line 193: "(not shown)". In fact, you do show the differences between ACI and BASE in Fig. 2.

13. line 214: According to Fig. 3b, the correlation coefficient in 0.78, not 0.40.

14. lines 215–216: The more strongly negative ACI−BASE precipitation differences in Central Europe associated with high PMratio events are a curious result. Why is the ratio of PM2.5/PM10 more important than PM2.5 alone? In general, at least in this region, I would expect that particles with diameter $< 2.5\,\mu$m are much more important than larger particles, especially for CCN and usually also for the aerosol direct radiative effects, because of their much larger number concentration. A somewhat remote possibility is that this result is related to giant aerosols enhancing precipitation, and thereby opposing the effect of smaller aerosols (this could be checked by looking at events defined wrt. to the difference PM10−PM2.5). Another possibility is that the result is coincidental, that is, more related to the different meteorological conditions associated with high vs. low values of the PMratio, rather than to the impact of aerosols on cloud microphysics. This risk is enhanced by the fact that all seasons, with different precipitation formation mechanisms, are lumped together.

15. lines 217–220: Why would the greater amount of small particles lead to reduced low cloudiness? Note that according to Fig. 6(d,e), the reduction in low clouds seems to be related mostly to the aerosol direct (and possibly semidirect) radiative effects rather than their effect on cloud microphysics.

16. line 236: "(significant differences)". Please refer to Fig. 2b to make it easier for the reader.

17. lines 237-240, 248-249: Given the very spatially scattered distribution of Region 3, it is hard to believe that this cluster really represents physically meaningful results, in spite of the apparent statistical significance. It seems more likely that the cluster analysis has just picked separately a group of points with increased frequency of large precipitation amounts, even if this increase itself might be caused by internal climate variability (i.e., be random). Note that grid points belonging to Region 3 are often neighboured by grid points belonging to Regions in which the frequency of heavy precipitation actually decreases.

18. lines 251–262: You should consider the statistical significance of the differences also in the case of Fig. 6. Some of the details discussed in this paragraph might not be robust.

19. line 270: "Zone 5" should be "Zone 4" (or "Region 4").

20. lines 304-305. It is not clear to what this sentence refers to. Please explain better, or remove.

21. Fig. 2: Note that in statistical testing, one should be aware of the risk of false positives. If a test is conducted at the significance level $p$=0.05, on average 5% of grid points will show "significant" differences, even if the differences between the two fields are actually random. It would be good to compute the fraction of significant differences and show it e.g. in the figure titles (it seems not to be much larger than 5% visually?). A more rigorous technique for looking at this would be "controlling the false discovery rate", see Wilks et al. (2016):

Wilks, D.S., 2016: "The stippling shows statistically significant grid points": How research results are routinely overstated and overinterpreted, and what to do about it. Bull. Amer. Meteor. Soc., 97, 2263–2273, *https://doi.org/10.1175/BAMS-D-15-00267.1*.

22. Consider marking the statistically significant differences also in Fig. 6.

**Technical and language corrections**

1. line 9: do you mean "time-mean spatially averaged"?

2. line 11: this should be "precipitation intensity regimes".

3. line 69: "dispersion" probably refers to "scattering"?

4. lines 73, 282, 285, 302 and 310. The use of "color" for describing clouds or aerosols is not clear, and certainly not standard scientific terminology. In the present context, "optical properties" would perhaps be the best term; for aerosols, "refractive index" could also be used.

5. line 159: replace "on a non-regular basis" with "in a non-linear scale".

6. line 256: add "causes" before "a reduction".

7. lines 277-279: The last sentence of Section 3 is not clear. Do you mean that in high PM10 conditions, clouds are preferentially located in the southern part of the area?

8. line 302: replace "order of magnitude ..." with "quantitatively this improvement is small".

9. line 310: replace "competence of CCN" with "efficiency of CCN".

10. In Figure 3, it is impossible to see black numbers plotted on black or dark blue background. Also, the units of the color bar should be % (not "score") in panels (c) and (d).

11. Caption of Fig. 4. The series used as the basis of the cluster analysis are not "time series" (in a time series, you would have time on the $x$-axis; here you have the precipitation threshold).

12. In Fig. 5, "Zona" is Spanish. "Zone" or "Region" would be English.

---

## Referee Comment (RC2) · Anonymous Referee #2 · 8 Jun 2020

General Comments

This paper shows results from 20 year run with regional climate model WRF-chem. Experiment setup includes simulations with different aerosol interaction. One clear conclusion of this paper is that both ACI and ARI lead to decrease of precipitation in Europe. Aerosols regional climate effects are still very uncertain and authors have carried out valuable simulations to increase our knowledge of aerosols regions effect on precipitation. Main question of this papper is what is the role of ACI and ARI in regional precipitation observations. However, i find some major comments on authods study. This paper is in scope of ACP and i recomend i to be publics after major revisions.

[Figure]

Major comments

1. Author clearly list list their findings on how ARI and ACI affects on rainy days, overall precipitation and low clouds. In figures term CLL is not opened, however in text this is indicated as low clouds. Text should mention what aerosol-cloud processes are included in the simulations, direct,indirect,semi-indirect, how these depend on aerosol type. How the aerosols it self formed in these experiments?

2. It's unclear was there simulation where both ACI and ARI were included. Author mention that there are areas where ACI and ARI effects cancel each other out. However due to non-linearities of aerosol-cloud effects, this conclusion would benefit from additional simulation where both ACI and ARI are included.

3. Also basic aerosols effect information should be shown, radiative forcing, direct and indirect. This helps reader to better understand the real effect of aerosols.

4. Only uncertainty regarding the model here is the aerosol setup. What is the role of model uncertainty? Example how much base case precipitation changes differs if you have slightly different initial condition in the model?

5. ARI simulations are not discussed except in figure 6. Similar analysis should be made also for ARI as done for ACI. I highly recomend also showing the results for ARI simulations.

6. In conclusion paper says that aerosol both decrese or increase precipitation , here it should also be stated why and where, what are the mechanisms causing these changes based on these simulations. Example in line 313 author says that decrese of precipitation is due to decrease of rainy days. What causes the decrease of rainy days?

7. Model aerosol configuration should be explained clearly, what natural and athropogenic aerosols are included.

Minor comments

Figure text in fgure 2. I suggest changes letters to beging of each sentence. (Top row) (a) Relative differences for precipitation between ACI and BASE experiments; (b)number of days of precipitation>0.1mm ; (c) and low clouds. Squares indicate points whose differences are significant for a p-value of 0.05.

In abstract line 9 spatially averaged should also mention the spatial region of the simulations which is the averages.

In method section I would recomend to include model section to describe the model it self

In line 91. Author states "In the BASE experiment aerosols are not treated interactively…. " Is this meaning that aerosol itself develops from vapors or aerosols are interaction with clouds?

In line 131. "). The simulations were run splitting the full period into sub-periods of 5years with a spin-up period of 4 months," this is unclear what has been done?

in line 134, "The evolution of greenhouse gases CO2, CH4and N2O were considered in accordance with the recommendation of Jerez et al. (2018)." This should be opened and expaliend the Jerez et.al paper

in line 150, "the relative differences.." relative to what?

in line 151 they refer tern "criteria" is unclear what criteria.

in line 160, clustering method used should be mentioned.

Titles in figure 5 should be changes to clusters. Also results in figure 4 and 5 should be discussed more. Figure 5 is somewhat puzzling.

---

## Author Comment (AC1) · 11 Sep 2020

**1   General Comments**

This paper documents relatively long (20 years) simulations of precipitation with the regional WRF-Chem model driven by ERA20C reanalysis.  Two simulations with interactive aerosols (one including only the aerosol direct radiative effect, another also the effect on cloud microphysics) are compared with a baseline simulation with fixed aerosols. It is found that the use of interactive aerosols decreases precipitation in Central/Eastern Europe and increases it in the Eastern Mediterranean. Detailed analyses

regarding the number of days with precipitation with different thresholds are carried out.

The treatment of aerosols in regional climate models is often rather primitive, and therefore I think the authors have carried out a valuable set of experiments. At the same time, I cannot recommend the publication of this paper in ACP, unless substantial improvements are made in the analysis and reporting of the results. The reasons for my concerns are outlined below.

We strongly appreciate the positive view of the reviewer and acknowledge the time devoted to the revision the manuscript and the fruitful comments leading to the improvement of the manuscript.

**2 Major comments**

1. My primary concern regarding this paper is that while it documents in some detail how precipitation changes, the physical interpretation of the results is rather lacking. The paper fails to properly address the question, what are the physical mechanisms leading to these changes in precipitation. Only a few cursory statements are made in this respect. The changes in precipitation could be caused by several mechanisms. They could arise through the impact of aerosols on cloud microphysics, or their impact on surface temperature (which could suppress convection), or through changes in large-scale meteorology (although the latter are probably small due to the use of nudging in the outer model domain).

   The revised version of the manuscript extends the discussion on the causes behind the changes, together with the analysis of additional meteorological variables as temperature, radiation and three-dimensional fields (included as supplementary materials and discussed within the text). Additionally, the introduction

has been extended in order to further include a description of the interactions leading to modifications in the precipitation regimes. Nonetheless, most of the studies on the current topic available in the scientific literature are case studies or ideal cases, for a certain type of cloud, type of aerosol or meteorological situation (see for example Khain et al (2007)). The aim of our work covers a climatic period and hence the separation of different circumstances is complex due to the internal variability of the model (the inner domain is large enough to generate it) and the mixture of aerosols and situations, in fact we obtain a important decrease of the temporal correlation among the experiments in some parts of the domain. Hence, the analysis presented in the manuscript focuses in statistical changes both in total precipitation and precipitation regimes. In the revised version of the manuscript we have deepened in the discussion of physical processes based on the available scientific literature and on the climate conditions of different European target areas.

2. To make it easier for the reader to interpret the findings, simulation results should be shown for additional physical quantities. It is very difficult to understand precipitation by looking at precipitation (and low clouds) alone. Most obviously, the paper should start with briefly showing how the aerosol fields (AOD, CCN, and aerosol radiative forcing, if available) differ from the baseline simulation, since these differences are the root cause for the changes in precipitation. I realize that some of this information is probably available in the cited papers by Palacios-Pena et al., but this paper should be able to stand alone — it should not be the reader's task to hunt for necessary information in other papers. Furthermore, changes in surface temperature are potentially important for convection, and they are referred to at a couple of occasions, but it would be better to actually show them. Other quantities that should be checked (and possibly shown, if their changes seem important for precipitation) include meteorological fields like surface pressure, relative humidity and mid-troposphere vertical velocity.

As commented for Item 1, the revised version of the manuscript extends the discussion on the causes behind the changes, together with the analysis of additional meteorological variables as temperature, radiation and three-dimensional fields. Most of the fields represented are added as Supplementary Material in order not to modify largely the structure of the manuscript.

3. The interpretation of the results is also complicated by the fact that data for all seasons are lumped together. Yet, the processes generating precipitation, and potentially their sensitivity to aerosols, depend strongly on the season. Especially concerning central-eastern Europe, which shows the clearest signal in precipitation, convective precipitation dominates in summer, while stratiform precipitation associated with synoptic weather systems dominates in winter. I recommend that the authors first look at precipitation on a season-to-season basis (at least distinguishing between the warm and cold seasons), and then focus the detailed analysis on the season(s) with the most meaningful signals.

   In a preliminary analysis, the seasonal interpretation was conducted. However, the most significant signals where depicted for the entire year, probably due simply because of statistical issued. Therefore, in the original manuscript we decided to represent only the annual results. However, following the Reviewer's advice, we present the seasonal analysis of the results, focusing mainly in the analysis of the differences between the simulations. Those analysis are presented as Supplementary Material and the results discussed along the text in the revised version of the manuscript.

4. While the authors have conducted both ARI and ACI simulations, the ARI results are not discussed much, except for Fig. 6. I strongly recommend to show the ARI-BASE and ACI-ARI differences, at least for the time-average precipitation in Fig. 2. It is vital information for understanding to which extent the precipitation changes arise from aerosol direct and indirect effects.

The reviewer is absolutely right. For that, the revised version of the manuscript includes the ARI simulations in the panels of figures. The text has been modified accordingly in order to discuss the new results.

5. There are rather many issues with the use of English language. At the end of the review, I list cases which I found disturbing for correct understanding of the text. This is not intended to serve as a complete language check.

We really appreciate your contribution to the improvement of the language. The final version of the manuscript will be revised by a native English speaker.

**3 Detailed comments**

1. line 16: should this be "eastern Mediterranean"?

   Yes. It has been corrected as suggested.

2. line 29: Can you add a reference to a publication listing the WCRP five major scientific challenges?

   We got this information from the web page of the World Climate Research Programme (WCRP). Checking it again we noted that that page has not been updated from a long time. We decided to remove this sentence from the manuscript.

3. line 32: I suggest replacing "The main tool" with "One of the main tools". The IPCC AR5 estimates of aerosol radiative forcing use satellite observations to adjust model- based results.

   Changed as suggested.

4. lines 40–44: A key point of the convective invigoration mechanism of Rosenfeld et al. (2008) is that the slower cloud-droplet-to-rain conversion allows the droplets
to be transported above the freezing level, and therefore, the latent heat released in freezing makes the convection more intense.

Following the Reviewer's advice, this point has been added to the revised version of the manuscript in the introduction and also is used in the discussion.

5. lines 46–49: It would be useful to give a bit more information on the cited studies (e.g., which regions were considered?).

We have incorporated further information about the state-of-the-art studies cited as well as some more new works, including area, aerosol type and size, etc..

6. lines 59–60: "and abundant number of cloud condensation nuclei (CCN) (Forkel et al., 2015) high enough for clouds to form without this variable being a limited factor". In fact, the lack of CCN is almost never a limiting factor for cloud formation (this could perhaps happen in remote marine locations in very specific conditions). However, a low CCN value may result in clouds that precipitate more readily, which can reduce the cloud lifetime and therefore the average cloud fraction.

Thanks for your comment. We have incorporated it to the revised version of the manuscript.

7. line 67: "black anthropogenic aerosols". Do you mean black carbon, or absorbing anthropogenic aerosols in more general? Furthermore, this paragraph gives the impression that anthropogenic aerosols cause warming and natural aerosols cause cooling, which is misleading. Many anthropogenic aerosols, most prominently sulfates, are largely non-absorbing, so the total effect of anthropogenic aerosols is probably one of radiative cooling.

The reviewer is right. We refer to black carbon as it is mainly generated by anthropogenic activity. We have clarified this point in the new version of the manuscript.

8. lines 112, 116: You mention the use of both the Goddard shortwave radiation scheme and the RRTMG scheme. To my knowledge, these are different radiation schemes. Please explain.

   The reviewer is right and the information was mistaken. We used the RRTMG scheme. This correction has been incorporated in the revised version of the manuscript.

9. lines 127-129: While AOD (it should be "aerosol optical depth") has been evaluated by Palacios-Pena et al. (2020), it would be definitely good to show the time-mean AOD fields also in this paper (see major comment 1).

   AOD fields and the differences among the experiments has been included in the revised version of the manuscript as supplementary material.

10. line 163: correlation matrix of what?

    The correlation matrix of the constructed series for each point. The constructed series are the differences between the number of days of precipitation for several thresholds. The sentence has been revised accordingly for the sake of clarity.

11. 11. lines 174-179: The spatial redistribution of precipitation is interesting, but is very difficult to figure out why it is happening, based on the information given in this paper. Please see the major comments 1-3.

    In order to provide clearer information the new figures included in the manuscript are presented and a deeper discussion is included.

12. 12. line 193: "(not shown)". In fact, you do show the differences between ACI and BASE in Fig. 2.

    We showh ACI-BASE but we do not present ARI-BASE. Anyway, new figure 2 of the manuscript is shown, and the differences commented.

13. 13. line 214: According to Fig. 3b, the correlation coefficient in 0.78, not 0.40.

We made a mistake here. We mean "In the case of PM10 ....... This paragraph has been rewritten including the 0.78 value for AOD and 0.4 for PM10.

14. 14. lines 215–216: The more strongly negative ACI-BASE precipitation differences in Central Europe associated with high PMratio events are a curious result. Why is the ratio of PM2.5/PM10 more important than PM2.5 alone? In general, at least in this region, I would expect that particles with diameter $<$ 2.5 $\mu$m are much more important than larger particles, especially for CCN and usually also for the aerosol direct radiative effects, because of their much larger number concentration. A somewhat remote possibility is that this result is related to giant aerosols enhancing precipitation, and thereby opposing the effect of smaller aerosols (this could be checked by looking at events defined wrt. to the difference PM10-PM2.5). Another possibility is that the result is coincidental, that is, more related to the different meteorological conditions associated with high vs. low values of the PMratio, rather than to the impact of aerosols on cloud microphysics. This risk is enhanced by the fact that all seasons, with different precipitation formation mechanisms, are lumped together.

We really appreciate this comment. We have been revising some papers about the role of Giant Aerosols by Feingold et al. and this could be key point that helps us to improve our explanation on the decrease of precipitation (amount and number of days) in that area as well as the increase in the eastern Mediterranean. Some more disscusion will be added to the new version of the manuscript about these processes.

15. lines 217–220: Why would the greater amount of small particles lead to reduced low cloudiness? Note that according to Fig. 6(d,e), the reduction in low clouds seems to be related mostly to the aerosol direct (and possibly semidirect) radiative effects rather than their effect on cloud microphysics.
We agree with the reviewer that the reduction in low clouds is related to the aerosol direct and semidirect radiative effects. But, the reduction of low clouds in ACI is larger than in ARI, therefore the role of microphysics could be important. In fact, an analysis performed similar to the one presented in Figure 3 shows how that this relationship exists. Anyway, we understand that our explanation is not complete. As mentioned before, some more plots including ARI experiment results have been added, as well as a much more extended explanation linking the reduction/increase of low clouds and precipitation based on both experiments (ARI and ACI) to direct, semidirect, and indirect effects.

16. 16. line 236: "(significant differences)". Please refer to Fig. 2b to make it easier for the reader.

    The reference to figure has been added.

17. lines 237-240, 248-249: Given the very spatially scattered distribution of Region 3, it is hard to believe that this cluster really represents physically meaningful results, in spite of the apparent statistical significance. It seems more likely that the cluster analysis has just picked separately a group of points with increased frequency of large precipitation amounts, even if this increase itself might be caused by internal climate variability (i.e., be random). Note that grid points belonging to Region 3 are often neighboured by grid points belonging to Regions in which the frequency of heavy precipitation actually decreases.

    We agree that Region 3 has no spatial structure. We perfectly understand the doubts of the reviewer about the physical meaning of this region. However, from the statistical point of view, we obtain that a important portion of grid cells presents an coherent increase of moderate and intense precipitation events. At the same time, we can found in the literature that this increase is supported by some physical processes. We think that it is important to keep the message, but at the same time to warm the reader about the need of deeper studies about

that, since it could be an artifact of the statistical methodology used.. We have rewritten the description of the behaviour of Region 3 trying to send the above message.

18. lines 251–262: You should consider the statistical significance of the differences also in the case of Fig. 6. Some of the details discussed in this paragraph might not be robust.

As commented before, we have now included all plots showing ACI and ARI experimented. In addition we have add the statistical significance as in figure 2, and when discussing the results we take into account the statistical significance.

19. line 270: "Zone 5" should be "Zone 4" (or "Region 4").

It has been fixed up.

20. 20. lines 304-305. It is not clear to what this sentence refers to. Please explain better, or remove.

We have removed that sentence. It do not provide any important message.

21. Fig. 2: Note that in statistical testing, one should be aware of the risk of false positives. If a test is conducted at the significance level p=0.05, on average 5% of grid points will show "significant" differences, even if the differences between the two fields are actually random. It would be good to compute the fraction of significant differences and show it e.g. in the figure titles (it seems not to be much larger than 5% visually?). A more rigorous technique for looking at this would be "controlling the false discovery rate", see Wilks et al. (2016): Wilks, D.S., 2016: "The stippling shows statistically significant grid points": How research results are routinely overstated and overinterpreted, and what to do about it. Bull. Amer. Meteor. Soc., 97, 2263–2273, https://doi.org/10.1175/BAMS-D-15- 00267.1.

We really appreciate the suggestion of the reviewer. We have calculated the fraction of significant differences and we refer them along the text. Anyway, we fix our attention on *significant areas* (group of nearby significant points ) that are far of being false positive as stated in Wilks et al 2016.

22. Consider marking the statistically significant differences also in Fig. 6.

    Thanks for your advice. As mentioned above, all maps of differences show the statistical significance.

**4  Technical and language corrections**

1. line 9: do you mean "time-mean spatially averaged"?

   Yes. Corrected.

2. line 11: this should be "precipitation intensity regimes".

   Corrected.

3. line 69: "dispersion" probably refers to "scattering"?

   Yes. Corrected

4. lines 73, 282, 285, 302 and 310. The use of "color" for describing clouds or aerosols is not clear, and certainly not standard scientific terminology. In the present context, "optical properties" would perhaps be the best term; for aerosols, "refractive index" could also be used.

   We acknowledge your suggestion, now we use *optical properties*

5. line 159: replace "on a non-regular basis" with "in a non-linear scale".

   Done.

6. line 256: add "causes" before "a reduction".

   Done.

7. lines 277-279: The last sentence of Section 3 is not clear. Do you mean that in high PM10 conditions, clouds are preferentially located in the southern part of the area?

   We mean that high load of PM10 are usually associated with synoptical conditions that transport the PM10 (dust) from the south. We have rewritten the sentence in order to make it clearer.

8. line 302: replace "order of magnitude ..." with "quantitatively this improvement is small".

   Done

9. line 310: replace "competence of CCN" with "efficiency of CCN".

   Done

10. In Figure 3, it is impossible to see black numbers plotted on black or dark blue background. Also, the units of the color bar should be % (not "score") in panels (c) and (d).

    Done

11. Caption of Fig. 4. The series used as the basis of the cluster analysis are not "time series" (in a time series, you would have time on the x-axis; here you have the precipitation threshold).

    Rigth. Now the caption reads .." Cluster analysis of rainy days: each color depicts a cluster with a different behavior of the ACI-BASE difference in number of days of precipitation over a threshold ...... "..

12. In Fig. 5, "Zona" is Spanish. "Zone" or "Region" would be English.

Fixed

---

## Author Comment (AC2) · 11 Sep 2020

**1   Main comments**

This paper shows results from 20 year run with regional climate model WRF-chem. Experiment setup includes simulations with different aerosol interaction.  One clear conclusion of this paper is that both ACI and ARI lead to decrease of precipitation in Europe.  Aerosols regional climate effects are still very uncertain and authors have carried out valuable simulations to increase our knowledge of aerosols regions effect on precipitation.  Main question of this paper is what is the role of ACI and ARI in

regional precipitation observations. However, I find some major comments on authors study. This paper is in scope of ACP and I recommend it to be published after major revisions.

We strongly appreciate the very positive and constructive comments of the reviewer and kindly acknowledge the time devoted to the revision of the manuscript. Please find below an item-by-item response to the Reviewer's ♯2 comments.

**2 Major comments**

1. Authors clearly list their findings on how ARI and ACI affects on rainy days, overall precipitation and low clouds. In figures term CLL is not opened, however in text this is indicated as low clouds. Text should mention what aerosol-cloud processes are included in the simulations, direct, indirect, semi-indirect, how these depend on aerosol type. How the aerosols itself formed in these experiments?

   CLL stands Clouds at Low Levels. The definition of this abbreviation has been added to the revised version of the manuscript. We agree with the Reviewer that the definition of the processes included in the different experiments lacks some detail. In the revised version of the manuscript, the Section devoted to the description of the experiments has been widely extended. Here, detailed descriptions of the processes involved in each experiment and the differences among them have been included. Basically, the BASE experiment does not include interactive aerosols. The ARI experiment introduces the aerosol-radiation interactions and the ACI experiments adds the aerosol interactions with the microphysics (aerosol-cloud interactions) in addition to the ARI simulations. Moreover, we have added some text explaining the origin and the formation of the different types of aerosols in the simulations. Basically, natural aerosols are generated by the interactions of atmospheric conditions with the land characteristics (vegetation, soil moisture, composition, etc.). Anthropogenic emissions of aerosols are taken from the ACCMIP initiative (Lamarque et al., 2010), as stated in the revised version of the manuscript.

2. It's unclear was there simulation where both ACI and ARI were included. Authors mention that there are areas where ACI and ARI effects cancel each other out. However due to non-linearities of aerosol-cloud effects, this conclusion would benefit from additional simulation where both ACI and ARI are included.

   As previously stated, the revised version of the manuscript includes a more detailed description of the experiments, where the issues raised by the Reviewer have been clarified. The ARI experiment includes only the aerosol-radiation interactions (mainly direct effects); in addition, the ACI experiments includes both the interactions of aerosols with radiation and with the cloud microphysics (indirect effects).

3. Also basic aerosols effect information should be shown, radiative forcing, direct and indirect. This helps reader to better understand the real effect of aerosols.

   We thank the Reviewer for his/her comment. In the revised version of the manuscript the results of all the experiments are shown regarding different aerosol-related variables, like AOD, PM10, PM2.5 and PMratio. Undoubtedly, this will help the reader to better understand the processes involved. Moreover, some complementary information has been added regarding the seasonal cycle of these variables

4. Only uncertainty regarding the model here is the aerosol setup. What is the role of model uncertainty? Example how much base case precipitation changes differs if you have slightly different initial condition in the model?

   The Reviewer raises a good point. Evidently, the internal variability plays an important role. In previous works of the research group (e.g. Jerez et al., 2020) the

role of the model initialization has been widely studied. However, in the revised version of the manuscript we analyze the impact of the aerosols on precipitation on a climatological scale. All the simulations have been identically initialized starting from the same chunks composing the different numerical experiments. We have to start from the hypothesis that the differences between the simulations come from the effects of the aerosols and their different treatment (only aerosol-radiation interactions or adding aerosol-cloud interactions). These aforementioned differences will be related both with direct, semi-direct and indirect effects, and their interaction with the internal variability of the model. Running new experiments analyzing that effect is unaffordable from a computational point of view at this time. In addition, the scientific literature consulted points to a negligible influence of the internal variability in this kind of experiments. On the other hand, the analysis conduced searches for the relationship between the changes obtained with the different concentration of different types of aerosols. In this analysis we include the statistical significance, so that we can corroborate the differences that can depart from the mere internal variability.

5. ARI simulations are not discussed except in Figure 6. Similar analysis should be made also for ARI as done for ACI. I highly recommend also showing the results for ARI simulations.

In the original version of the manuscript we decided to include only the ARI analysis when the differences between the simulations were caused essentially by the changes induced by the microphysics of the model. This was initially done in order to minimize the number of Figures and the length of the text. Nevertheless, we fully understand the Reviewer's concern. The revised version of the manuscript includes the analysis of the differences of the fields obtained both for ARI and ACI experiments.

6. 6. In conclusion paper says that aerosol both decrease or increase precipitation , here it should also be stated why and where, what are the mechanisms causing

these changes based on these simulations. Example in line 313 author says that decrease of precipitation is due to decrease of rainy days. What causes the decrease of rainy days?

The scientific literature that covers the topic of the effects of aerosols on precipitation -and the physical processes involved- focus mainly on study cases. The objective of the work includes the analysis of changes in precipitation, amount and regimes, together with its relationship with different types of aerosols from a climatological perspective. This approach slightly hampers the direct association to physical processes, because the effects of aerosols depend on the meteorological situation, the type of aerosols, and in our case of the differences in the time evolution. The straightforwards effect produced can evolve in time and space indirectly due to the internal variability of the model, since simulations do not use nudging in the inner domain and simulations are transient (continuous). The statistical analysis carried out shows how diverse areas respond differently to the aerosol feedbacks. While in some areas precipitation is reduced when including aerosol interactions (Central Europe), this impact is low for total precipitation. However, if we focus in the number of rainy days, this impact is noticeable, affecting days with less precipitation. Conversely, in the Mediterranean the response of precipitation is the contrary, and the type of aerosols and the environmental conditions also differs. Therefore, we understand that the physical explanations of the results found are not fully included in the manuscript; however, in the revised version, this discussion about physical processes has been extended based on the results from other studies. As an example, Khain et al. (2008) indicate the high variability of the changes in precipitation due to modifications in the type of aerosols and environmental conditions.

7. Model aerosol configuration should be explained clearly, what natural and anthropogenic aerosols are included.

As aforementioned, the Methodology section has now included a detailed description on the setup of the experiments and the aerosols involved in the simulations.

**3 Minor comments**

1. Figure text in figure 2. I suggest changes letters to beging of each sentence. (Toprow) (a) Relative differences for precipitation between ACI and BASE experiments;(b)number of days of precipitation>0.1mm ; (c) and low clouds. Squares indicate points whose differences are significant for a p-value of 0.05.

   Done as suggested.

2. In abstract line 9 spatially averaged should also mention the spatial region of the simulations which is the averages.

   The averages are estimated over the whole domain. Done as suggested.

3. In method section I would recommend to include model section to describe the model itself

   As mentioned before, a much more detailed description of model and experimental setup has been added to the manuscript.

4. In line 91. Author states "In the BASE experiment aerosols are not treated interactively.... " Is this meaning that aerosol itself develops from vapors or aerosols are interaction with clouds?

   This section has been modified. In the BASE experiment aerosol properties affecting the physics of the model are constant in space and time (for radiation, AOD; and for microphysics, the cloud condensation nucleii are constant).

5. In line 131. "). The simulations were run splitting the full period into sub-periods of 5 years with a spin-up period of 4 months," this is unclear what has been done?

The total period simulated for each experiment (BASE, ARI and ACI) is of 20 years. Instead of doing a run of 20 years long, we split each simulation in 4 chunks of 5 years with an spin-up period of 4 months. This spin-up time is removed and the 4 chunks are pasted. This is done following the recommendation of Jerez et al. (2020) in order to make experiments faster.

6. In line 134, "The evolution of greenhouse gases CO2, CH4 and N2O were considered in accordance with the recommendation of Jerez et al. (2018)." This should be opened and explained the Jerez et.al paper

   Done as suggested.

7. In line 150, "the relative differences.." relative to what?

   The relative differences are calculated as the differences among the experiments (ACI-BASE) divided by the BASE case and multiplied by 100, therefore relative to the BASE case.

8. In line 151 they refer tern "criteria" is unclear what criteria.

   The criteria defined in the above paragraph, the intensity and and extension over the defined thresholds. It has been clarified in the new version of the manuscript.

9. In line 160, clustering method used should be mentioned.

   The clustering method is composed by several steps, the final one is the K-means method. This has been clarified in the text.

10. Titles in figure 5 should be changed to clusters. Also results in figure 4 and 5 should be discussed more. Figure 5 is somewhat puzzling.

    As suggested, some more discussion has been added and zones are renamed as clusters in Figure 5.

---

## Referee Report (RR1)

**General comments**

I thank the authors for their substantial efforts in revising the manuscript, and for the additional material included.

Based on this material, it is clear that there are multiple reasons for the differences in precipitation between the different experiments. These include not only the "straightforward" effects of ARI and ACI, but also changes in atmospheric circulation (whether caused by the different treatment of aerosols, internal variability, or their combination). So it is understandable that in many cases the physical reasons for the precipitation differences remain unclear. Some of the physical explanations provided by the authors seem reasonable, while some appear uncertain or even unlikely. A few such instances are pointed out in the specific comments below. In addition, there are rather many minor technical/editorial issues that should be corrected to make the paper easier to read. The current manuscript gives an impression that you did not have quite enough time for a proper proof-reading in the end.

**Note:** The comments and line numbers in this review are based on the manuscript version "acp-2020-381-manuscript-version3.pdf". This version seems to differ in some details from the marked-up manuscript provided as part of the author response.

**Specific comments**

1. line 45: Please indicate the region studied (the title of the Da Silva et al. (2018) paper suggests it was the Euro-Mediterranean region).

2. line 60: I suggest deleting "high enough for clouds to form without this variable being a limiting factor".

3. line 185: For completeness, also mention the ARI−BASE differences in spatially-averaged total precipitation. Presumably small?

4. lines 198–199: Replace "being stronger in winter" with "the absolute changes being largest in winter"? (This is not true for the relative changes shown in Figs. 7 and Fig. 9 in the Supplementary material).

5. line 201: presumably, this should be "decrease of clouds".

6. lines 202 and 203: "changes in temperature are opposite for tasmax and tasmin ...". Are they? Based on Figs. 14 and 15 in the Supplementary material, the spatial correlation between the changes in maximum and minimum temperatures could even be positive.

7. lines 251–257: If I understand the logic of the reasoning here correctly, it is suggested that a greater concentration of small particles acts to reduce low clouds and precipitation due to semi-direct effects (i.e., black carbon aerosols). However, it seems rather treacherous to draw such a conclusion based on the PMratio alone. The mass fraction of aerosols below 2.5 $\mu$m diameter does not necessarily tell much about BC. Since BC fields are available in the model (Supplementary Fig. 5), you could check this hypothesis more directly by adding an analysis wrt. "BC events" in Fig. 4.

Note that other explanations are also possible. For example, it could also be that cases with a lot of small particles happen to be associated atmospheric circulation types with drier-than-average atmospheric conditions in this region (obviously, this is speculation too).

8. line 301: "... higher concetration of small particles that modifies the properties of clouds, inhibiting precipitation processes again by reducing clouds due to microphysics processes, since [in] this area there is a prevalence of small aerosols". Again, it is not obvious that the prevalence of small aerosols should lead to reduced low cloudiness (what would be the microphysical process causing this?). Rather this looks like a case where some factor X (possibly changes in atmospheric circulation) reduces both cloudiness and precipitation. Perhaps it would be better to say that the explanation is not clear, rather than guess.

9. lines 306–307. "While small particles inhibit the formation of clouds by semidirect effects, larger aerosols ease the cloud formation and precipitation by indirect effects"? Referring to comment 8, the role of semidirect effects could be better checked by looking at the BC concentration rather than PM2.5 or the PMratio. The PMratio does not tell much about BC. It is also not clear why larger aerosol ease the cloud formation (the possible role of giant aerosols on precipitation is another matter). Note that the lower size limit for CCN is around 0.1 $\mu$m. So if you draw the line between "small" and "large" aerosol particles at a 2.5 $\mu$m diameter, then the vast majority of CCN are "small'. Perhaps this paragraph could be deleted altogether?

10. lines 370–371: "Our experiments show a extra depletion of cloudiness, probably related to a faster evaporation of drops". Replace "probably" with "possibly"? Very little actual evidence for this has been shown in the paper.

11. lines 377–386: Would it be worth adding something like "It would be interesting to see to which extent other regional models would reproduce the current results for the Euro-CORDEX region"? As a friendly reminder of the possible (and probable) model-dependence of the results...

12. Fig. 7: I think it would be most logical to show this figure already in the early part of the paper, i.e. between Figs. 1 and 2.

**Technical corrections**

1. line 125: replace "departing" with "originating".

2. line 136: Mention the figure numbers in the Supplementary material to make it easier for the reader to locate these figures. Also in other instances where Supplementary material is referred to (although perhaps not on lines 151–154).

3. line 163: Add parenthesis around ARCI−BASE.

4. line 186: The text refers to Fig. 2a, but the figure panels are not identified by letters in Fig. 2. Please also check for other instances like this in the text.

5. lines 221 and 251: Figure number is missing.

6. lines 264–290, and Figs. 6 and 8. Different terms are used in the text ("Region") and in the figures ("Cluster" in Fig. 6, "Zone" in Fig. 8). A consistent notation would be preferable — I would vote for "Cluster", since this is based on cluster analysis.

7. line 295: Figure 4 should be Figure 2?

8. line 304: Why "whole" ARI effect?

9. line 308: Should "Figure 8" be "Figure 7"?

10. line 312: "Figure 7" should be "Figure 8".

11. line 368 and 370: "CNN" shoud be "CCN"

12. Figure 2: In the title of the middle column, "BAS" should be "BASE".

13. Figure 3: Panel titles would be needed here. Now it is not clear which quantities are shown. Also, mention in the caption that the squares refer to significant differences *between the ACRI and BASE experiments*.

14. Caption of Fig. 4. Replace "non-constant linear behavior" with "non-linear behavior".

15. Fig. 5: The cluster numbers 1-5 should be shown.

16. Caption of Fig. 7: Please correct the units for PM10.

17. Figure 8: Please indicate which experiments are compared here.

18. In the Supplementary material, please name figures as S1, S2 and so on (note that the remaining comments use this notation).

19. Regarding the 3rd column in the figures in Supplementary material, the figure captions should indicate whether the absolute or relative differences are shown. This seems to vary, and now the reader has to figure it out on a case-by-case basis.

20. In Fig. S1, the title of the 3rd column should be "ARCI$-$ARI".

21. In Figs. S6–S22, replace "BAS" with "BASE".

22. Caption of Fig. S4: replace "UNIDADES" with "unitless".

23. Caption of Fig. S5. What is "BC2"? At which height is it defined?

24. Fig. S8: Are these results really given in "number of days per month" as the caption indicates? The numerical values (up to 10–20) seem very large.

25. Caption of Fig. S10: What are the units? Percentage points?

26. Caption of Fig. S17: The numerical values seem to be in units of "Pa", not "mb" as the caption states.

27. Figures S21 and S22: Please check the units. Temperature differences of several K seem very large (and much larger than those at the surface level in Figs. S14 and S15).

28. Please have the language checked. I noted rather many issues with the English language, especially in the new parts of the manuscript.

---

## Author Response (AR2)

**Response to Reviewer ♯2.**
**Precipitation response to Aerosol-Radiation and Aerosol-Cloud Interactions in Regional Climate Simulations over Europe**

JP. Montávez

**General comments**

I thank the authors for their substantial efforts in revising the manuscript, and for the additional material included.

Based on this material, it is clear that there are multiple reasons for the differ ences in precipitation between the different experiments. These include not only the "straightforward" effects of ARI and ACI, but also changes in atmospheric cir culation (whether caused by the different treatment of aerosols, internal variability, or their combination). So it is understandable that in many cases the physical reasons for the precipitation differences remain unclear. Some of the physical explanations provided by the authors seem reasonable, while some appear uncertain or even unlikely. A few such instances are pointed out in the specific comments below. In addition, there are rather many minor technical/editorial issues that should be corrected to make the paper easier to read. The current manuscript gives an impression that you did not have quite enough time for a proper proof-reading in the end.

We strongly appreciate the very detailed and useful revision provided by the reviewer. Without a doubt, the reviewer has greatly helped to improve the quality of the manuscript. We apologize for the large number of mistakes along the manuscript and for not providing a proper proof-reading version. In the new version of the manuscript we follow all the suggestion of the reviewer and treat to answer the remaining questions about the physical explanations of the processes.

We send the answer to all questions, specific and technical. The version we send now, has been also sent to a native English for a grammar revision. Sorry for not providing the english corrected version at this moment, but it was impossible to receive before the dead line for submitting the answer.

**Specific comments**

1. line 45: Please indicate the region studied (the title of the Da Silva et al. (2018) paper suggests it was the Euro-Mediterranean region).

   Yes, it has been added.

2. line 60: I suggest deleting "high enough for clouds to form without this variable being a limiting factor".

   It has been deleted

3. line 185: For completeness, also mention the ARI-BASE differences in spatially-averaged total precipitation. Presumably small?

   It has been added. The differences are still smaller (0.1%)

4. lines 198–199: Replace "being stronger in winter" with "the absolute changes being largest in winter"? (This is not true for the relative changes shown in Figs. 7 and Fig. 9 in the Supplementary material).

   It has been changed

5. line 201: presumably, this should be "decrease of clouds".

   Yes, the reviewer is right. It has been corrected.

6. lines 202 and 203: "changes in temperature are opposite for tasmax and tasmin ...". Are they? Based on Figs. 14 and 15 in the Supplementary material, the spatial correlation between the changes in maximum and minimum temperatures could even be positive.

   The reviewer is right. The spatial correlations of changes in tmax and tmin are always positive (although small). We have rewritten the sentence. The new sentence is: *Changes in temperature are different for*

tasmax *and* tasmin. *They are larger for* tasmax, *specilly in ARCI, reaching differences around 0.5K and presenting spatial patterns quite similar to those of CLL, While* tasmin *does not present any correlation with CLL.*

7. lines 251–257: If I understand the logic of the reasoning here correctly, it is suggested that a greater concentration of small particles acts to reduce low clouds and precipitation due to semi-direct effects (i.e., black carbon aerosols). However, it seems rather treacherous to draw such a conclusion based on the PMratio alone.The mass fraction of aerosols below 2.5 $\mu$ m diameter does not necessarily tell much about BC. Since BC fields are available in the model (Supplementary Fig. 5), you could check this hypothesis more directly by adding an analysis wrt. BC events in Fig. 4. Note that other explanations are also possible. For example, it could also be that cases with a lot of small particles happen to be associated atmospheric circulation types with drier-than-average atmospheric conditions in this region (obviously, this is speculation too).

Our results show that PM2.5 is able to explain some of the changes in precipitation, if fact selecting some intense episodes the differences are large. However the response of NDP (Number of days of precipitation) to the intensity and extension of episodes for PMratio, is quite high, leading to a very significant decrease of the days in precipitation when such episodes are selected. The analysis of BC do not present any significant signal. The physical explanation we found revising the literature is that while small particles inhibit precipitation (semidirect effects and indirect effects), larger particles enhance precipitation. Therefore, as our results show, the largest NDP reduction appears when small particles dominate. On the other hand, as the reviewer points, these cases could be associated to some atmospheric circulation types that transport particles from far areas. This also could explain the no significant role of BC. This is a work that we plan to do during the next months.

8. line 301: "... higher concetration of small particles that modifies the properties of clouds, inhibiting precipitation processes again by reducing clouds due to microphysics processes, since [in] this area there is a prevalence of small aerosols". Again, it is not obvious that the prevalence of small aerosols should lead to reduced low cloudiness (what would be the microphysical process causing this?). Rather this looks like a case where some factor X (possibly changes in atmospheric circulation) reduces both cloudiness and precipitation. Perhaps it would be better to say that the explanation is not clear, rather than guess.

We understand the reviewer comment. The text has been rewritten according to the reviewer sugesstion, showing that this explanation is just a possible cause and adding other indirect effects related to changes in circulation.

9. lines 306–307. "While small particles inhibit the formation of clouds by semidirect effects, larger aerosols ease the cloud formation and precipitation by indirect effects"? Referring to comment 8, the role of semidirect effects could be better checked by looking at the BC concentration rather than PM2.5 or the PMratio. The PMratio does not tell much about BC. It is also not clear why larger aerosol ease the cloud formation (the possible role of giant aerosols on precipitation is another matter). Note that the lower size limit for CCN is around 0.1 $\mu$m. So if you draw the line between "small" and "large" aerosol particles at a 2.5 $\mu$m diameter, then the vast majority of CCN are "small'. Perhaps this paragraph could be deleted altogether?

The full paragraph has been deleted. Probably it is quite speculative.

10. lines 370–371: "Our experiments show a extra depletion of cloudiness, probably related to a faster evaporation of drops". Replace "probably" with "possibly"? Very little actual evidence for this has been shown in the paper.

It has been changed

11. lines 377–386: Would it be worth adding something like "It would be interesting to see to which extent other regional models would reproduce the current results for the Euro-CORDEX region"? As a friendly reminder of the possible (and probable) model-dependence of the results. . .

A new sentence has been added in order to include the suggestion of extending this study to other models in order to investigate the model dependence of the results.

12. Fig. 7: I think it would be most logical to show this figure already in the early part of the paper, i.e. between Figs. 1 and 2.

Yes, figure 7 has been reordered.

**1 Technical corrections**

1. line 125: replace "departing" with "originating".

   Done

2. line 136: Mention the figure numbers in the Supplementary material to make it easier for the reader to locate these figures. Also in other instances where Supplementary material is referred to (although perhaps not on lines 151–154).

   We have included references to figures of the Suplementary Material.

3. line 163: Add parenthesis around ARCI-BASE.

   Done

4. line 186: The text refers to Fig. 2a, but the figure panels are not identified by letters in Fig. 2. Please also check for other instances like this in the text.

   Done

5. lines 221 and 251: Figure number is missing.

   All figure numbers has been revised.

6. 6. lines 264–290, and Figs. 6 and 8. Different terms are used in the text ("Region") and in the figures ("Cluster" in Fig. 6, "Zone" in Fig. 8). A consistent notation would be preferable — I would vote for "Cluster", since this is based on cluster analysis.

   Done, we use cluster in all cases.

7. . line 295: Figure 4 should be Figure 2?

   Done. REPASAR TODAS LAS FIGURAS

8. line 304: Why "whole" ARI effect?

   This paragraph was removed.

9. line 308: Should "Figure 8" be "Figure 7"?

   This paragraph was removed.

10. line 312: "Figure 7" should be "Figure 8".

    Fixed.

11. line 368 and 370: "CNN" shoud be "CCN"

    Changed

12. Figure 2: In the title of the middle column, "BAS" should be "BASE".

    Fixed

13. Figure 3: Panel titles would be needed here. Now it is not clear which quantities are shown. Also, mention in the caption that the squares refer to significant differences between the ARCI and BASE experiments.

    Figure 3 has been redone. Now, all information is included in de caption.

14. Caption of Fig. 4. Replace "non-constant linear behavior" with "non-linear behavior".

    Done

15. Fig. 5: The cluster numbers 1-5 should be shown.

    Fixed.

16. Caption of Fig. 7: Please correct the units for PM10.

    Done

17. Figure 8: Please indicate which experiments are compared here.

    Done. (ARCI-BASE)

18. In the Supplementary material, please name figures as S1, S2 and so on (note that the remaining comments use this notation).

    Done

19. Regarding the 3rd column in the figures in Supplementary material, the figure captions should indicate whether the absolute or relative differences are shown. This seems to vary, and now the reader has to figure it out on a case-by-case basis.

    When the differences are relative it is indicated in the caption. Some times the differences are multiplied by a factor. This has been now indicated in the captions.

20. In Fig. S1, the title of the 3rd column should be "ARCI-ARI".

    Fixed

21. In Figs. S6–S22, replace "BAS" with "BASE".

    Done

22. Caption of Fig. S4: replace "UNIDADES" with "unitless".

    Done

23. Caption of Fig. S5. What is "BC2"? At which height is it defined?

    BC2 is hydrophilic Black Carbon (BC2) $(10^2 \mu g/kg - dryair)$ at level 7 of model mesh ( 1000m). It has been included in the caption

24. Fig. S8: Are these results really given in "number of days per month" as the caption indicates? The numerical values (up to 10–20) seem very large.

    Yes, the results are given in number of days per year. It has been corrected.

25. Caption of Fig. S10: What are the units? Percentage points?

    The cloud coverage is given as percentage. It has been pointed out in the figure caption.

26. Caption of Fig. S17: The numerical values seem to be in units of "Pa", not "mb" as the caption states.

    Yes, it has been corrected.

27. Figures S21 and S22: Please check the units. Temperature differences of several K seem very large (and much larger than those at the surface level in Figs. S14 and S15).

    Yes, temperature differences are multiplied by 10, for a better reading f the color scale. It has been corrected in the figure caption.

28. Please have the language checked. I noted rather many issues with the English language, especially in the new parts of the manuscript.

    We have sent the final version to an native English reviewer for language correction.

[revised manuscript text omitted]
), specially during autumn and springtime, leading to a reduction of clouds and precipitation. Although the explanation in not clear, a possible explanation would be that ARCI experiment enhances this effect by the higher concentration of small particles modifying the properties of the clouds, inhibiting precipitation processes again by reducing clouds due to microphysical processes, since over this area there is a prevalence of small aerosols (see PMratio in Figure 3 and Figures S1-S5 of Supplementary Material).

On the other hand, there are areas where the effects of ARI and ACI tend to cancel each other , or have different effects on small or large rainfall. This is the case of the area of Balkans, where the ARI effect tends to decrease precipitation, while ACI tend to increase rainfall, being the net effect (ARCI) negligible (Figure 2). This behavior can be attributed 
[revised manuscript text omitted]

---

## Author Response (AR3)

**Final comments.**
**Precipitation response to Aerosol-Radiation and Aerosol-Cloud Interactions in Regional Climate Simulations over Europe**

JP. Montávez

**General comments**

Dear Editor,

First of all we would like to thanks again the very detailed and useful revision provided by the reviewers. Here we include the tracked changes of the manuscript after the corrections of a native englisth reviewer.

Best regards

Juan Pedro Montávez.

[revised manuscript text omitted]